# Mirage or Method? How Model–Task Alignment Induces Divergent RL Conclusions

**Haoze Wu**[1]* **Cheng Wang**[2]* **Wenshuo Zhao**[1] **Junxian He**[3]
[1]Zhejiang University [2]National University of Singapore [3]HKUST
waithz@zuaa.zju.edu.cn wangcheng@u.nus.edu junxianh@cse.ust.hk

## Abstract

Recent advances in applying reinforcement learning (RL) to large language models (LLMs) have led to substantial progress. In particular, a series of remarkable yet often counterintuitive phenomena have been reported in LLMs, exhibiting patterns not typically observed in traditional RL settings. For example, notable claims include that a single training example can match the performance achieved with an entire dataset, that the reward signal does not need to be very accurate, and that training solely with negative samples can match or even surpass sophisticated reward-based methods. However, the precise conditions under which these observations hold—and, critically, when they fail—remain unclear. In this work, we identify a key factor that differentiates RL observations: whether the pretrained model already exhibits strong *Model-Task Alignment*, as measured by pass@k accuracy on the evaluated task. Through a systematic and comprehensive examination of a series of counterintuitive claims, supported by rigorous experimental validation across different model architectures and task domains, our findings show that while standard RL training remains consistently robust across settings, many of these counterintuitive results arise only when the model and task already exhibit strong model-task alignment. In contrast, these techniques fail to drive substantial learning in more challenging regimes, where standard RL methods remain effective. Code is available at https://github.com/hkust-nlp/model-task-align-rl.

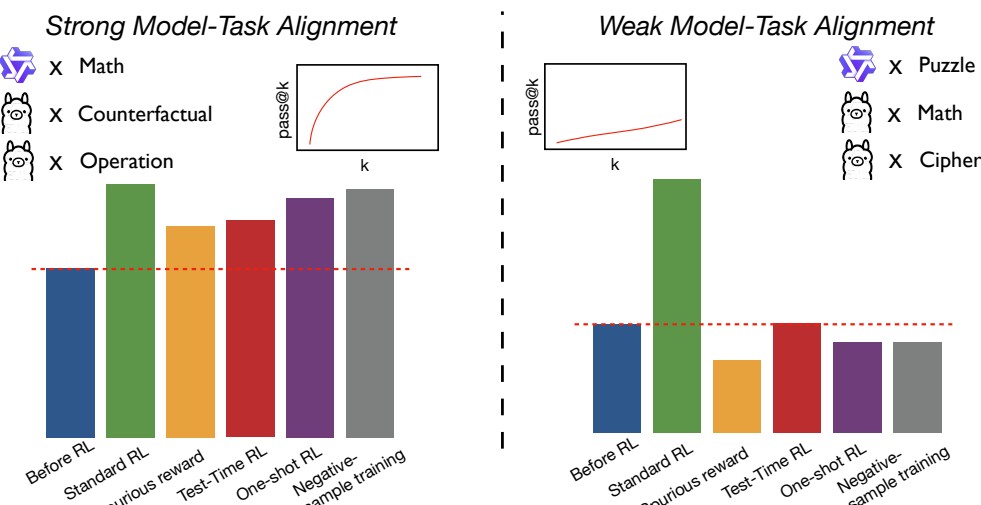

Figure 1: Model-task alignment, which is measured by pass@k accuracy on the evaluated task, drives distinct outcomes from the same series of RL approaches.

---

*Equal Contribution. Work done during visit to HKUST.

## 1 INTRODUCTION

Reinforcement Learning (RL) (Sutton et al., 1998) has emerged as a transformative post-training technique for Large Language Models (LLMs), enabling them to follow instructions (Ouyang et al., 2022) and align with human preferences (Ziegler et al., 2019; Rafailov et al., 2024). A particularly prominent application focuses on enhancing reasoning capabilities, as exemplified by breakthrough models such as OpenAI-o1 (Jaech et al., 2024), DeepSeek-R1 (Guo et al., 2025), QwQ (Team, 2025), and Kimi-1.5 (Team et al., 2025). These systems demonstrate remarkable performance across reasoning-intensive domains including coding (Jain et al., 2024), mathematics (Lewkowycz et al., 2022; He et al., 2024), and logical reasoning (Liu et al., 2025; Chen et al., 2025).

While RL yields significant performance improvements in LLM reasoning—mirroring the success of RL in traditional domains such as games (Silver et al., 2017a;b)—we also observe several remarkable yet often counterintuitive empirical phenomena. These effects appear to be unique to LLMs and would be considered unexpected in traditional RL settings. For instance, single training examples can match or rival full-dataset training performance (Wang et al., 2025), ground-truth reward may be surprisingly dispensable (Shao et al., 2025), and training with negative samples alone can match sophisticated reward-based methods (Agarwal et al., 2025).

Although these findings have generated considerable enthusiasm, the precise conditions under which they hold, and when they break down, remain insufficiently explored. Given that these observations may have important implications for RL practices, it is concerning that the conclusions are largely based on limited experimental settings, where Qwen models (Qwen et al., 2025) trained on mathematical tasks dominate the landscape.

To this end, we carry out a systematic empirical investigation of several notable RL claims, supported by rigorous experimental validation across diverse model architectures and task domains. Concretely, we experiment with both Qwen and non-Qwen models on math and other tasks. Our controlled experiments reveal that *model-task alignment*, defined as the degree to which model capabilities match task requirements, is a critical indicator for categorizing RL observations. Specifically, models benefit from noisy rewards, test-time RL (Zuo et al., 2025), minimal training, and negative-sample training primarily within their domains of expertise, where these techniques fail for unfamiliar tasks even though standard RL training can succeed. Interestingly, we also observe that certain meta-patterns hold consistently across different settings. For instance, one-shot RL training is generally effective for the specific task to which the training example belongs, and negative-sample training helps stabilize model entropy, even though it does not always lead to overall improvements in accuracy.

We evaluate the "alignment" between model capabilities and task requirements using pass@k accuracy, which we find to be a reliable indicator for distinguishing these counterintuitive RL phenomena. Our hypothesis is that strong, inherent model capabilities can be readily activated through minimal training, even when guided by incorrect reward signals, whereas unfamiliar tasks demand substantially more effort—cases that we argue dominate when scaling up RL compute. Concurrent work (Wu et al., 2025) investigates the mechanism behind spurious rewards and attributes their effectiveness primarily to data leakage in Qwen models on the test set. However, our results suggest otherwise: we find that spurious rewards remain effective even in the absence of contamination, provided the model already exhibits strong alignment on the evaluated task.

Our study reveals that, unlike traditional RL training, distinct RL mechanisms emerge in the context of LLMs, depending on whether the pretrained model is already familiar with the target tasks. On the one hand, this suggests that RL phenomena should be interpreted with extra caution, as they may only reflect one of these two mechanisms. On the other hand, it also opens up opportunities for jointly optimizing base model pretraining (or mid-training) and RL post-training. For example, one might enhance the domain-specific capabilities of the base model during mid-training, enabling effective RL with limited training data and potentially inaccurate reward signals, or alternatively, allocate most compute resources to the RL stage using carefully curated training data and precise reward signals.

## 2 ON UNIQUE PHENOMENA OF RL TRAINING IN LLM REASONING

Reinforcement Learning from Verifiable Rewards (RLVR) has achieved significant success in improving language model reasoning. While similar gains in accuracy from standard training have

also been observed in traditional RL domains such as games, we have noticed several phenomena that appear unique to LLMs and would not typically be expected in conventional settings. For example, we highlight several remarkable, and at times counterintuitive, observations below: **(a) Unexpected robustness to unreliable or absent rewards:** Shao et al. (2025) demonstrate that random and incorrect reward signals can improve model performance, while Agarwal et al. (2025) show that reward-free, entropy-minimization objectives can rival reward-based approaches. Test-Time Reinforcement Learning (TTRL) proposed by Zuo et al. (2025) further reinforces this trend by generating reward signals through aggregating majority-vote outcomes, thereby guiding the model to evolve itself on the test set. Together, these suggest surprising fault tolerance in RL training that challenges standard assumptions about the critical role of accurate reward signals. **(b) One-shot training sufficiency:** Wang et al. (2025) report that training on a single carefully selected example can match or exceed performance from full dataset training, challenging assumptions about data volume requirements. **(c) Negative-only signal effectiveness:** Zhu et al. (2025) demonstrate that using exclusively negative reward signals achieves comparable results to standard RL training while maintaining beneficial entropy properties.

These findings carry significant implications. If broadly confirmed, they would necessitate shifts in resource allocation—such as prioritizing data selection algorithms over dataset scale, questioning the necessity of highly accurate reward modeling, and potentially introducing new research directions. Therefore, we believe it is important to assess whether these conclusions hold in general, and if not, under what conditions they succeed or fail. Clarifying these patterns would not only help us understand the limitations of the current findings but also, in the opposite direction, reveal new opportunities for modifying models so that these findings become valid, thereby making RL training substantially easier. In this work, we will investigate these observations through controlled experiments comprehensively.

## 2.1 CENTRAL HYPOTHESIS: MODEL-TASK ALIGNMENT DEPENDENCY

As most of the findings discussed above are based on mathematical reasoning tasks using Qwen models (Qwen et al., 2025; Yang et al., 2025), a natural question arises: do these results generalize to other settings? For instance, Shao et al. (2025) reported that spurious rewards were ineffective with Llama (Meta, 2024) models on mathematical tasks. However, we argue that treating Qwen+math as merely a special case is an overly superficial categorization. It remains unclear what specifically makes Qwen+math unique, and what the deeper, more essential factors might be. We propose a guiding hypothesis for designing and categorizing experimental settings, which we call **Model-Task Alignment Dependency**: *the effectiveness of these unique RL findings fundamentally depends on the degree of alignment between a model's inherent capabilities and the requirements of the task domain.* In other words, they depend on the model's proficiency on the evaluated task. This hypothesis may or may not hold, but we will use it as a framework to categorize experimental settings in terms of whether the model–task combination is aligned or misaligned.

**Quantifying Model-Task Alignment with pass@k.** To systematically evaluate the degree of alignment between a model's inherent capabilities and the requirements of specific task domains, we employ the pass@k metric as our primary measure of model-task proficiency. Pass@k represents the probability that at least one correct solution appears among k independent samples generated by the model for a given problem. This metric effectively captures how well a model's existing knowledge and reasoning patterns align with the demands of a particular task.

Formally, for a problem $x_i$ from evaluation dataset $\mathcal{D}$, we generate $n$ samples ($n \geq k$) and count the number of correct samples as $c_i$, then the unbiased estimator of pass@k over the dataset is: $\text{pass@k} := \mathbb{E}_{x_i \sim \mathcal{D}} \left[ 1 - \binom{n-c_i}{k} / \binom{n}{k} \right]$.

## 2.2 STRATEGIC MODEL AND TASK SELECTION

Building on our *Model-Task Alignment Dependency* hypothesis outlined in Section 2.1, we strategically design model-task combinations that test the boundaries of current claims in RL for language model reasoning. Our experimental design is motivated by the critical need to distinguish between findings that represent universal RL properties versus those that emerge from specific model-task capability alignments. We evaluate two representative language models from different families: Qwen2.5-7B-Base (Qwen et al., 2025) and Llama-3.1-8B-Instruct (Meta, 2024), enabling system-

atic comparison across model architectures with varying baseline capabilities while controlling for architectural differences at comparable parameter scales.

Our evaluation encompasses mathematical and logical reasoning domains. For mathematical reasoning, we employ AIME24 (AIME, 2024), MATH500 (Hendrycks et al., 2021) and AMC23 (AMC, 2023). For logical reasoning, we utilize SynLogic (Liu et al., 2025) (synthetic puzzles with 35 task types, we use the validation split), BBH (Suzgun et al., 2022) (multi-step reasoning tasks), BBEH Kazemi et al. (2025) (extended-difficulty version), and KOR-Bench (Ma et al., 2024) (knowledge-orthogonal reasoning across five categories).

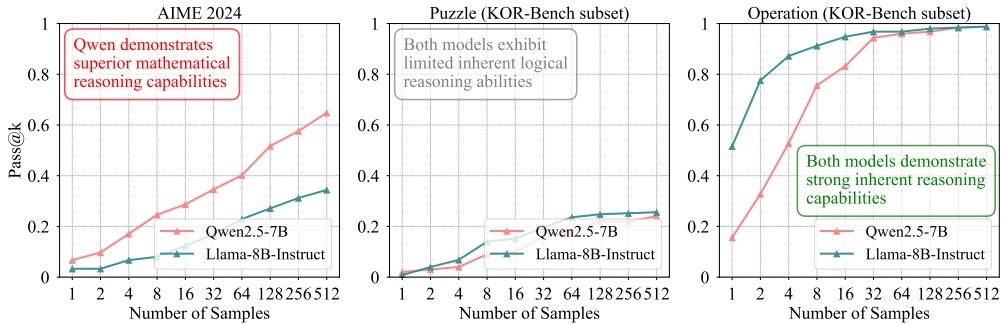

Figure 2: Pass@k for different tasks. Different LLMs have significantly different abilities on different tasks, which will affect how the RL techniques perform across model-task combinations.

To operationalize our hypothesis, we systematically measure alignment strength using pass@k metrics across all model-task combinations. As demonstrated in Figure 2, models exhibit markedly different inherent capabilities across domains. Based on comprehensive evaluation (full results in Appendix C), we identify cases of strong model-task alignment, such as Qwen2.5 on mathematical domains and both models on Operation and Counterfactual subsets of KOR-Bench, as well as weak model-task alignment cases, such as Llama3.1 on mathematical domains and both models on other logical reasoning tasks. This categorization enables us to test whether counterintuitive RL phenomena are artifacts of specific model-task alignments or represent fundamental properties of reinforcement learning in language model reasoning.

## 2.3 THE CONTAMINATION HYPOTHESIS

Concurrent work from Wu et al. (2025) proposed an alternative hypothesis, where they specifically focus on the spurious reward pattern and suggest that it stems primarily from dataset contamination during pre-training. They further confirmed the presence of data leakage in the Qwen models on several mathematical benchmarks. While we acknowledge contamination as a valid concern, our hypothesis diverges by emphasizing the distinction between contamination and inherent task proficiency. In particular, models may demonstrate strong task performance without direct contamination of the test data. In what follows, we categorize different experimental settings based on their contamination and inherent model-task alignment status. Later, in our experiments, we will empirically show that contamination is not the underlying cause; instead, model-task alignment serves as a more reliable differentiator.

To verify our hypothesis, we extend contamination analysis beyond Qwen-Math combinations. Following Wu et al. (2025), we evaluate model generation given partial prompts while preserving word boundaries (more details are provided in Appendix D). We employ greedy decoding and calculate both exact match (EM) rates and ROUGE-L scores, where ROUGE-L scores of 1.0 indicate perfect reconstruction. Table 1 alongside Appendix D show that Operation and Counterfactual subsets have no contamination, yet both models demonstrate strong inherent reasoning capabilities with high pass@k scores (see Appendix C). As we will show in our experiments, contamination is not the necessary condition for the effectiveness of these RL phenomena. Based on our contamination analysis and pass@k measurements, we categorize experimental settings into three groups: **Red (Potential Contamination + Strong Model-Task Alignment)**: Qwen2.5 on mathematical domains. **Gray (No Contamination + Weak Model-Task Alignment)**: Llama3.1 on mathematical domains; both models on SynLogic, BBH, BBEH, and Logic, Cipher, Puzzle subsets of KOR-Bench. **Green (No**

| Model | Portion | AMC 23 | | MATH500 | | Puzzle | | Operation | |
|---|---|---|---|---|---|---|---|---|---|
| | | ROUGE | EM | ROUGE | EM | ROUGE | EM | ROUGE | EM |
| **Qwen2.5-7B** | 0.4 | 63.78 | 23.91 | 50.36 | 8.20 | 19.56 | 0.00 | 21.37 | 0.00 |
| | 0.6 | 64.42 | 33.73 | 60.98 | 21.20 | 19.62 | 0.00 | 24.25 | 0.00 |
| | 0.8 | 73.23 | 49.39 | 66.42 | 40.20 | 19.24 | 0.00 | 20.18 | 0.00 |
| **Llama-3.1-8B** | 0.4 | 27.18 | 0.00 | 23.09 | 0.60 | 18.27 | 0.00 | 21.83 | 0.00 |
| | 0.6 | 30.64 | 0.00 | 40.56 | 3.80 | 17.31 | 0.00 | 18.34 | 0.00 |
| | 0.8 | 44.54 | 4.81 | 48.33 | 17.8 | 15.85 | 0.00 | 16.75 | 0.00 |

Table 1: Contamination Analysis across model-task combinations. Portion refers to the truncation ratio of the prompt used to test whether models can complete the remaining content. **Red** indicates potential contamination with strong model-task alignment; **Gray** indicates no contamination with weak model-task alignment; **Green** indicates no contamination with strong model-task alignment.

**Contamination + Strong Model-Task Alignment)**: Both models on Operation and Counterfactual subsets of KOR-Bench.

## 3 EXPERIMENTAL SETUP

**Training Datasets and Evaluation.** Except for the experiments on Test-Time RL (Section 4.2), we use DeepScaleR (Luo et al., 2025) as the training set for mathematical tasks and the training split of SynLogic-Easy (Liu et al., 2025) for logical tasks. Evaluation datasets are as described in Section 2.2. Following SynLogic (Liu et al., 2025), all evaluations are conducted in a zero-shot setting, with avg@8 metrics computed for AIME 2024 and SynLogic to mitigate variance.

**Training Configuration.** Our experiments default to using the DAPO algorithm with a group size of 16. Its effectiveness has been demonstrated on math (Yu et al., 2025) and logic tasks (Liu et al., 2025). We set $\epsilon_{low} = 0.2, \epsilon_{high} = 0.28$, max prompt length $= 2048$, max generation length $= 8192$. We use dynamic sampling, which is crucial for improving the reward on SynLogic, and set max_num_gen_batches $= 2$. For logical tasks, each sampled batch often contains very few samples with non-zero reward variance. We apply two strategies: (1) if neither generated batch contains samples with non-zero reward variance, the second batch is used for training; (2) if the number of available samples is smaller than the batch size, samples are duplicated. We do not use a length penalty. In most experiments, we set $lr = 1e^{-6}$, batch size $= 128$, mini batch size $= 64$, temperature $= 1.0$. We fix all key hyperparameters across experiments to ensure that observed differences primarily reflect model–task alignment rather than tuning effort.

## 4 RQ1 – REWARD SIGNAL: HOW CRITICAL IS IT?

This section investigates the role of reward signal quality and its impact on RL performance for LLMs. Previous work in Reinforcement Learning with Human Feedback has shown that more accurate reward models do not always lead to better downstream performance (Chen et al., 2024). In the specific context of LLM reasoning, initial studies found that models with strong inherent reasoning abilities exhibit surprising robustness to noisy reward signals, whereas weaker models show poor noise tolerance (Lv et al., 2025; Shao et al., 2025). Building on these findings, we extend the analysis by considering diverse reward signals across different model-task combinations. Hyperparameters follow Section 3, and training runs for 300 steps (see Appendix B for more details).

### 4.1 RESULTS

We present results in Table 2. From the results, we identify three critical findings regarding the impact of reward signal quality on model performance (Appendix E provides additional discussion):

**Ground Truth Rewards Consistently Outperform All Alternatives.** Across both model families and all task domains, utilizing ground truth rewards consistently yields the highest performance improvements. For instance, Qwen2.5-7B achieves substantial gains on AIME24 (14.2 vs. baseline 3.3) and MATH500 (71.0 vs. baseline 40.8) when trained with accurate reward signals. This establishes ground truth rewards as the gold standard for RL training in reasoning tasks.

| | Math Tasks | | | Logic Tasks | | | | | | | |
| | AIME24 | MATH500 | AMC | SynLogic | BBH | BBEH | KOR Benchmark | | | | |
| | | | | | | | OP | CF | Puzzle | Logic | Cipher |
|---|---|---|---|---|---|---|---|---|---|---|---|
| **Qwen2.5-7B Family** | | | | | | | | | | | |
| Base | 3.3 | 40.8 | 31.0 | 1.5 | 45.2 | 1.2 | 27.2 | 17.2 | 0.8 | 8.0 | 4.8 |
| **RLVR (External Reward)** | | | | | | | | | | | |
| Correct | $14.2_{+10.9}$ | $71.0_{+30.2}$ | $62.4_{+31.4}$ | $42.6_{+41.1}$ | $62.7_{+17.5}$ | $6.8_{+5.6}$ | $82.4_{+55.2}$ | $79.6_{+62.4}$ | $16.8_{+10.0}$ | $46.4_{+38.4}$ | $20.4_{+15.6}$ |
| Random | $10.0_{+6.7}$ | $57.5_{+16.7}$ | $45.7_{+14.7}$ | $10.2_{+8.7}$ | $32.7_{-12.5}$ | $0.0_{-1.2}$ | $53.6_{+26.4}$ | $30.8_{+13.6}$ | $1.2_{+0.4}$ | $6.8_{-1.2}$ | $3.6_{-1.2}$ |
| Incorrect | $6.7_{+3.4}$ | $57.0_{+16.2}$ | $43.1_{+12.1}$ | $0.0_{-1.5}$ | $30.3_{-14.9}$ | $0.0_{-1.2}$ | $60.8_{+33.6}$ | $12.8_{-4.4}$ | $0.4_{-0.4}$ | $6.4_{-1.6}$ | $3.2_{-1.6}$ |
| Format | $6.7_{+3.4}$ | $55.3_{+14.5}$ | $48.9_{+17.9}$ | $1.5_{0.0}$ | $44.4_{-0.8}$ | $2.4_{+1.2}$ | $37.2_{+10.0}$ | $21.6_{+4.4}$ | $0.8_{0.0}$ | $6.8_{-1.2}$ | $4.4_{-0.4}$ |
| **Self-Rewarded Reinforcement Learning** | | | | | | | | | | | |
| Vote | $13.3_{+10.0}$ | $69.4_{+28.6}$ | $58.2_{+27.2}$ | $2.8_{+1.3}$ | $33.6_{-11.6}$ | $0.0_{-1.2}$ | $56.4_{+29.2}$ | $16.3_{-0.9}$ | $0.8_{0.0}$ | $6.8_{-1.2}$ | $3.2_{-1.6}$ |
| EM | $11.6_{+8.3}$ | $70.8_{+30.0}$ | $57.8_{+26.8}$ | $1.5_{0.0}$ | $37.5_{-7.7}$ | $0.0_{-1.2}$ | $67.2_{+40.0}$ | $27.2_{+10.0}$ | $0.8_{0.0}$ | $6.8_{-1.2}$ | $3.2_{-1.6}$ |
| **Llama3.1-8B-Instruct Family** | | | | | | | | | | | |
| Base | 3.3 | 32.5 | 20.2 | 0.8 | 38.6 | 4.1 | 60.4 | 86.4 | 2.0 | 28.8 | 8.4 |
| **RLVR (External Reward)** | | | | | | | | | | | |
| Correct | $6.7_{+3.4}$ | $38.6_{+6.1}$ | $25.1_{+4.9}$ | $21.0_{+20.2}$ | $49.1_{+10.5}$ | $4.3_{+0.2}$ | $76.0_{+15.6}$ | $88.8_{+2.4}$ | $15.6_{+13.6}$ | $34.4_{+7.6}$ | $11.6_{+3.2}$ |
| Random | $3.3_{0.0}$ | $26.8_{-5.7}$ | $21.3_{+1.1}$ | $0.0_{-0.8}$ | $32.1_{-6.5}$ | $4.1_{0.0}$ | $69.2_{+8.8}$ | $87.2_{+0.8}$ | $0.8_{-1.2}$ | $23.6_{-5.2}$ | $4.4_{-4.0}$ |
| Incorrect | $2.1_{-1.2}$ | $26.4_{-6.1}$ | $18.7_{-1.5}$ | $0.8_{0.0}$ | $30.2_{-8.4}$ | $3.8_{-0.3}$ | $70.0_{+9.6}$ | $83.2_{-3.2}$ | $0.8_{-1.2}$ | $19.2_{-9.6}$ | $4.4_{-4.0}$ |
| Format | $3.1_{-0.2}$ | $31.5_{-1.0}$ | $18.7_{-1.5}$ | $0.8_{0.0}$ | $36.4_{-2.2}$ | $4.1_{0.0}$ | $68.8_{+8.4}$ | $85.6_{-0.8}$ | $2.0_{0.0}$ | $28.0_{-0.8}$ | $6.4_{-2.0}$ |
| **Self-Rewarded Reinforcement Learning** | | | | | | | | | | | |
| Vote | $4.6_{+1.3}$ | $37.7_{+5.2}$ | $23.0_{+2.8}$ | $1.5_{+0.7}$ | $35.9_{-2.7}$ | $4.3_{+0.2}$ | $67.2_{+6.8}$ | $83.2_{-3.2}$ | $2.0_{0.0}$ | $28.0_{-0.8}$ | $8.8_{+0.4}$ |
| EM | $5.1_{+1.8}$ | $38.3_{+5.8}$ | $25.0_{+4.8}$ | $0.8_{0.0}$ | $34.8_{-3.8}$ | $4.1_{0.0}$ | $73.6_{+13.2}$ | $87.2_{+0.8}$ | $2.0_{0.0}$ | $23.6_{-5.2}$ | $7.6_{-0.8}$ |

Table 2: Comprehensive evaluation of different reward signals in RL. "Vote" denotes Majority Voting, "EM" means entropy minimization on self-generated samples only; OP: Operation ; CF: Counterfactual. **Red** indicates potential contamination with strong model-task alignment; **Gray** indicates no contamination with weak model-task alignment; **Green** indicates no contamination with strong model-task alignment.

**Model-Task Alignment Determines Robustness to Noisy Rewards.** The effectiveness of noisy reward signals depends on model-task alignment strength across our three experimental categories. In settings with strong alignment (Red and Green categories), models demonstrate surprising robustness to spurious rewards, with Qwen2.5-7B maintaining reasonable performance on mathematical tasks and both models showing improvements on Operation and Counterfactual tasks even with random rewards. Conversely, in weak alignment settings, spurious rewards consistently fail to provide meaningful improvements, as seen with Llama3.1-8B on mathematical tasks and both models on challenging logical reasoning benchmarks. This pattern confirms that alignment strength, rather than contamination alone, determines robustness to noisy rewards.

**Limited Effectiveness of Self-Rewarded Methods.** Self-Rewarded Reinforcement Learning methods, including majority voting and entropy minimization, consistently underperform compared to external reward-based approaches. While self-rewarded methods shows some promise on mathematical tasks for Qwen2.5-7B (majority vote achieves 69.4 on MATH500), it fails to match the performance of ground truth external rewards and shows poor generalization to logical reasoning tasks across both model families.

## 4.2 TEST-TIME RL

Test-Time Reinforcement Learning (TTRL) (Zuo et al., 2025) addresses a fundamental challenge in LLM development: how to improve model performance on unlabeled test data without access to ground-truth labels for reward signals. It prompts the model to generate multiple responses to each test question and use the most frequent answer as the label for reward signals. Although the model is trained on the unlabeled test set, this approach is essentially no different from Self-Rewarded Reinforcement Learning when majority voting is employed. Thus, we are also curious whether TTRL remains effective for different models and in domains beyond mathematics.

Table 3 shows the results of the Qwen and Llama models on different tasks. Due to the limited scale of the test dataset, we trained for 30 steps on all test datasets. It could be observed that in settings where the model–task alignment is strong, TTRL yields substantial improvements, as exemplified by

Qwen on math tasks and Operation subset. For tasks in which the model lacks initial prior knowledge, TTRL fails to deliver improvements or yields only marginal gains. As discussed by Zuo et al. (2025), majority voting is the foundation of TTRL. We also recorded the variation of Maj@16 during the training process; the results are shown in Table 4. We can observe that, in settings where TTRL yields substantial improvements, Maj@16 consistently rises throughout training. Especially for Qwen on Operation subset, it achieves an absolute gain of 16.4 points. This further underscores that TTRL's efficacy hinges on strong model–task alignment, rather than on contamination.

| Model | MATH500 | SynLogic | OP | Model | MATH500 | SynLogic | OP |
|---|---|---|---|---|---|---|---|
| Qwen2.5-7B | 40.8 | 1.5 | 27.2 | Llama-3.1-8B-Instruct | 32.5 | 0.8 | 60.4 |
| +TTRL | $62.1_{+21.3}$ | $1.8_{+0.3}$ | $55.6_{+28.4}$ | +TTRL | $41.2_{+8.7}$ | $0.8_{0.0}$ | $83.6_{+23.2}$ |

Table 3: Test-Time Reinforcement Learning (TTRL) performance changes. TTRL produces significant gains only when model-task alignment is strong (red and green cells).

| | Step 0 | Step 5 | Step 10 | Step 15 | Step 20 | Step 25 | Step 30 |
|---|---|---|---|---|---|---|---|
| Qwen+Math500 | 54.2 | 60.6 | 64.3 | 68.2 | 67.1 | 69.3 | 70.5+16.3 |
| Qwen+SynLogic | 2.2 | 3.0 | 3.7 | 4.4 | 4.4 | 4.4 | 5.2+3.0 |
| Qwen+OP | 46.0 | 53.6 | 55.6 | 57.2 | 58.8 | 60.0 | 60.0+16.4 |
| Llama+Math500 | 46.3 | 48.6 | 51.3 | 53.2 | 53.9 | 55.0 | 54.7+8.4 |
| Llama+SynLogic | 1.5 | 1.5 | 2.2 | 1.5 | 2.2 | 2.2 | 2.2+0.7 |
| Llama+OP | 73.6 | 78.0 | 79.6 | 84.0 | 83.6 | 86.8 | 88.4+14.8 |

Table 4: The variation of Maj@16 as training progresses. In tasks where TTRL brings significant improvements (red and green), Maj@16 continues to improve with training.

# 5 RQ2 – IS ONE-SHOT ENOUGH FOR RL TO WORK?

Wang et al. (2025) demonstrated that training on a single carefully selected question can yield performance comparable to full dataset training, challenging conventional assumptions about data volume requirements in RL. Wang et al. (2025) designs a selection algorithm based on the variance of training rewards, and we denote samples selected by this algorithm as $m_{selected}$ for mathematical tasks and $l_{selected}$ for logical tasks. In addition to that, we also randomly selected one or two samples from the dataset to form $(m_{random}, l_{random})$ and $(m'_{random}, l'_{random})$ for comparison. The specific examples we used are detailed in Appendix K. The remaining experimental settings are consistent with those described in Section 3, and we train models for 300 steps.

| Dataset | Math Tasks | | | Logic Tasks | | | | | | | |
|---|---|---|---|---|---|---|---|---|---|---|---|
| | | | | | | | KOR Benchmark | | | | |
| | AIME24 | MATH500 | AMC | SynLogic | BBH | BBEH | OP | CF | Puzzle | Logic | Cipher |
| Qwen2.5-7B | | | | | | | | | | | |
| ∅ | 3.3 | 40.8 | 31.0 | 1.5 | 45.2 | 1.2 | 27.2 | 17.2 | 0.8 | 8.0 | 4.8 |
| full set | $14.2_{+10.9}$ | $71.0_{+30.2}$ | $62.4_{+31.4}$ | $42.6_{+41.1}$ | $62.7_{+17.5}$ | $6.8_{+5.6}$ | $82.4_{+55.2}$ | $79.6_{+62.4}$ | $16.8_{+10.0}$ | $46.4_{+38.4}$ | $20.4_{+15.6}$ |
| random-1 | $10.7_{+7.4}$ | $58.7_{+17.9}$ | $53.1_{+22.1}$ | $0.8_{-0.7}$ | $40.2_{-5.0}$ | $0.0_{-1.2}$ | $60.4_{+33.2}$ | $36.8_{+19.6}$ | $0.8_{0.0}$ | $6.4_{-1.6}$ | $4.4_{-0.4}$ |
| random-2 | $12.5_{+9.2}$ | $63.0_{+22.2}$ | $55.7_{+22.7}$ | $2.4_{+0.9}$ | $43.1_{-2.1}$ | $1.2_{0.0}$ | $67.2_{+40.0}$ | $56.8_{+39.6}$ | $2.0_{+1.2}$ | $3.2_{-4.8}$ | $4.8_{0.0}$ |
| selected-1 | $12.3_{+9.0}$ | $65.2_{+24.4}$ | $55.2_{+24.2}$ | $0.8_{-0.7}$ | $39.9_{-5.3}$ | $0.0_{-1.2}$ | $69.2_{+42.0}$ | $38.4_{+21.2}$ | $0.8_{0.0}$ | $8.0_{0.0}$ | $6.4_{+1.6}$ |
| Llama3.1-8B-Instruct | | | | | | | | | | | |
| ∅ | 3.3 | 32.5 | 20.2 | 0.8 | 38.6 | 4.1 | 60.4 | 86.4 | 2.0 | 28.8 | 8.4 |
| full set | $6.7_{+3.4}$ | $38.6_{+6.1}$ | $25.1_{+4.9}$ | $21.0_{+20.2}$ | $49.1_{+10.5}$ | $4.3_{+0.2}$ | $76.0_{+15.6}$ | $88.8_{+2.4}$ | $15.6_{+13.6}$ | $34.4_{+7.6}$ | $11.6_{+3.2}$ |
| random-1 | $3.8_{+0.5}$ | $30.5_{-2.0}$ | $21.1_{+0.9}$ | $0.8_{0.0}$ | $35.1_{-3.5}$ | $3.8_{-0.3}$ | $73.6_{+13.2}$ | $85.6_{-0.8}$ | $1.2_{-0.8}$ | $28.0_{-0.8}$ | $8.8_{+0.4}$ |
| random-2 | $2.7_{-0.6}$ | $33.1_{+0.6}$ | $21.1_{+0.9}$ | $0.8_{0.0}$ | $36.7_{-1.9}$ | $4.1_{0.0}$ | $70.0_{+9.6}$ | $86.4_{0.0}$ | $2.8_{+0.8}$ | $27.2_{-1.6}$ | $8.4_{0.0}$ |
| selected-1 | $3.7_{+0.4}$ | $30.3_{-2.2}$ | $22.3_{+2.1}$ | $0.8_{0.0}$ | $34.4_{-4.2}$ | $3.8_{-0.3}$ | $69.2_{+8.8}$ | $88.8_{+2.4}$ | $2.0_{0.0}$ | $19.2_{-9.6}$ | $6.8_{-1.6}$ |

Table 5: One-shot RL Results. OP: Operation; CF: Counterfactual. We only observe the effectiveness of one-shot reinforcement learning in settings with strong model-task alignment (red and green).

## 5.1 RESULTS

We present results in Table 5. Based on the experimental results, we identify two critical findings regarding the effectiveness of one-shot reinforcement learning:

**One-shot RL Success Depends on Model-Task Alignment.** The effectiveness of one-shot reinforcement learning is highly contingent on the alignment between model capabilities and task domain requirements. In strong alignment settings (Red and Green categories), both models demonstrate remarkable ability to generalize from single examples: Qwen2.5-7B achieves performance comparable to full dataset training on mathematical tasks (MATH500: 65.2 vs. full training 71.0), while both Qwen2.5-7B and Llama3.1-8B-Instruct show substantial improvements on Operation and Counterfactual tasks (e.g., Llama on Operation: 69.2 vs. baseline 60.4). However, this success does not extend to weak alignment settings, where both models show minimal improvements across challenging logical reasoning benchmarks. This suggests that one-shot RL serves as an effective fine-tuning mechanism only when models already possess strong foundational capabilities.

**Sample Selection Strategy Shows Limited Impact.** Contrary to expectations, the sophisticated sample selection algorithm proposed by Wang et al. (2025). does not consistently outperform random sample selection. For Qwen2.5-7B on mathematical tasks, both selected and random samples achieve similar performance levels (MATH500: selected 65.2 vs. random 58.7 and 63.0), while for Llama3.1-8B-Instruct, the differences are negligible across all benchmarks. This finding challenges the assumption that reward variance-based selection provides substantial advantages over simpler random sampling approaches.

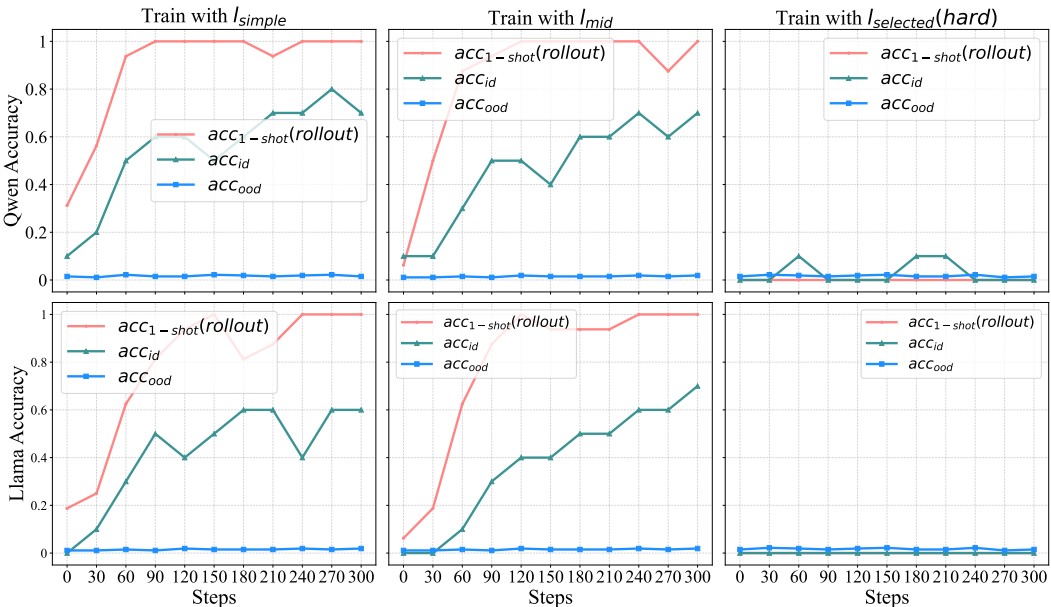

Figure 3: The changes in two models' accuracy during the training. If the initial rollout accuracy is non-zero, both models rapidly fit the employed samples ($l_{simple}$, $l_{mid}$) and exhibit generalization within the same subtask; however, we observe no generalization to puzzles of other types.

## 5.2 DISCUSSION

Wang et al. (2025) showed that training on a single sample for mathematical tasks can quickly improve the accuracy of that sample and also lead to improvements on the test set. We attempt to verify this conclusion on logical tasks. Considering that the initial rollout accuracy of the model on $l_{selected}$ is $0$, we additionally sample two examples whose initial rollout accuracies on Qwen2.5-7B are $5/16$ and $1/16$ (on Llama-3.1-8B-Instruct are $3/16$ and $1/16$), denoted as $l_{simple}$ and $l_{mid}$. During training, we track three metrics: the rollout accuracy of these examples $acc_{1-shot}$, the accuracy of the subtask to which this example belongs (in-distribution) $acc_{id}$, and the accuracy of other subtasks in SynLogic (out-of-distribution) $acc_{ood}$. The results are shown in Figure 3.

**One-shot RL possesses the ability to generalize within the distribution.** When the problem is relatively simple (with an initial rollout accuracy that is not zero), the model's rollout accuracy on that sample quickly increases. Although the initial rollout accuracy of $l_{mid}$ on Qwen is only one-fifth that of $l_{simple}$ (on Llama is one-third), it still attains a high rollout accuracy within a few dozen steps. Since GRPO and DAPO compute advantages via intra-group normalization, the model is unable to derive any informative feedback from samples whose initial rollout accuracy is zero. Moreover, we observe that the test accuracy for the same subtask also continues to improve, demonstrating effective within-distribution generalization.

**One-shot RL struggles to generalize to other types of logic puzzles.** We find that while models can improve on tasks similar to their training example, they fail to transfer learning to different puzzle types. This suggests that one-shot learning primarily exploits existing model capabilities rather than developing new reasoning skills.

# 6 RQ3 — DOES RL WORK WITH ONLY NEGATIVE SAMPLES?

Recent work (Zhu et al., 2025) has demonstrated that training exclusively on negative samples can be surprisingly effective for model reasoning. However, these findings are primarily observed in scenarios with strong model-task alignment. We investigate whether negative-only training generalizes to weak model-task alignment scenarios, where models lack strong foundational capabilities.

**Implementation Details.** In our implementation, negative Sample Reinforcement (NSR) masks out all trajectories with reward 1 (correct answers) when computing the policy gradient, leaving only negative-rewarded samples to drive updates. Conversely, Positive Sample Reinforcement (PSR) ignores trajectories with reward 0 and optimizes only on positively rewarded samples. All other hyperparameters remain identical to the DAPO baseline described in Section 3.

## 6.1 RESULTS

| | Math Tasks | | | Logic Tasks | | | | | | | |
|---|---|---|---|---|---|---|---|---|---|---|---|
| | | | | | | | KOR Benchmark | | | | |
| | AIME24 | MATH500 | AMC | SynLogic | BBH | BBEH | OP | CF | Puzzle | Logic | Cipher |
| Qwen2.5-7B | 3.3 | 40.8 | 31.0 | 1.5 | 45.2 | 1.2 | 27.2 | 17.2 | 0.8 | 8.0 | 4.8 |
| DAPO | $14.2_{+10.9}$ | $71.0_{+30.2}$ | $62.4_{+31.4}$ | $42.6_{+41.1}$ | $62.7_{+17.5}$ | $6.8_{+5.6}$ | $82.4_{+55.2}$ | $79.6_{+62.4}$ | $16.8_{+10.0}$ | $46.4_{+38.4}$ | $20.4_{+15.6}$ |
| NSR | $13.9_{+10.6}$ | $68.7_{+27.9}$ | $63.5_{+32.5}$ | $1.5_{0.0}$ | $41.2_{-4.0}$ | $1.6_{+0.4}$ | $60.4_{+33.2}$ | $36.8_{+19.6}$ | $2.0_{+1.2}$ | $6.8_{-1.2}$ | $4.8_{0.0}$ |
| PSR | $14.0_{+10.7}$ | $70.3_{+29.5}$ | $63.1_{+32.1}$ | $24.8_{+23.3}$ | $57.1_{+11.9}$ | $4.3_{+3.1}$ | $73.6_{+46.4}$ | $38.4_{+21.2}$ | $9.2_{+8.4}$ | $31.2_{+23.2}$ | $11.2_{+6.4}$ |
| Llama3.1-8B | 3.3 | 32.5 | 20.2 | 0.8 | 38.6 | 4.1 | 60.4 | 86.4 | 2.0 | 28.8 | 8.4 |
| DAPO | $6.7_{+3.4}$ | $38.6_{+6.1}$ | $25.1_{+4.9}$ | $21.0_{+20.2}$ | $49.1_{+10.5}$ | $4.3_{+0.2}$ | $76.0_{+15.6}$ | $88.8_{+2.4}$ | $15.6_{+13.6}$ | $34.4_{+7.6}$ | $11.6_{+3.2}$ |
| NSR | $7.9_{+4.6}$ | $36.9_{+4.4}$ | $24.7_{+4.5}$ | $0.0_{-0.8}$ | $34.2_{-4.4}$ | $4.3_{+0.2}$ | $67.2_{+6.8}$ | $86.4_{0.0}$ | $2.0_{0.0}$ | $28.0_{-0.8}$ | $5.2_{-3.2}$ |
| PSR | $7.9_{+4.6}$ | $35.7_{+4.2}$ | $23.6_{+3.4}$ | $13.0_{+11.5}$ | $43.3_{+4.7}$ | $4.1_{0.0}$ | $69.2_{+8.8}$ | $89.6_{+3.2}$ | $12.0_{+11.2}$ | $34.4_{+7.6}$ | $10.8_{+2.4}$ |

Table 6: Results of NSR and PSR under different settings. When Model-Task alignment is strong, both NSR and PSR yield pronounced performance gains for all models (Red and Green). Conversely, under weak alignment, NSR-trained models exhibit no noticeable improvement (Gray).

Table 6 summarizes the performance of NSR and PSR relative to the full-signal DAPO baseline across our three experimental categories. It reveals distinct patterns based on model-task alignment:

**Strong Model-Task Alignment Enables Effective Negative-Sample Learning.** In settings with strong model-task alignment (Red and Green categories), both NSR and PSR show comparable effectiveness, recovering most of the performance gains achieved by full-signal DAPO. For Qwen2.5-7B on mathematical tasks, both approaches achieve ~95% of the DAPO improvement (MATH500: NSR 68.7 and PSR 70.3 vs. DAPO 71.0). This demonstrates that when models already possess strong domain capabilities, either positive-only or negative-only signals can effectively drive learning.

**Weak Model-Task Alignment Reveals Superior Performance of Positive-Only Signals.** In weak alignment settings (Gray category), PSR consistently outperforms NSR across logical reasoning tasks. For instance, on SynLogic, PSR enables meaningful improvements (Qwen2.5-7B: 1.5 vs. 24.8, Llama3.1-8B: 0.8 vs. 13.0), while NSR shows minimal gains. Overall, while PSR and NSR

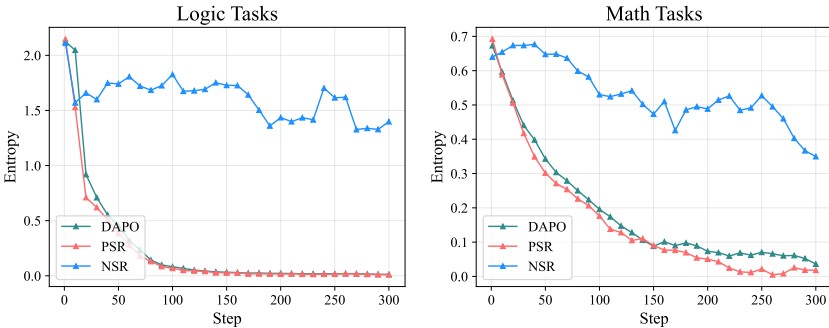

Figure 4: Entropy Dynamics of Qwen2.5-7B during Training. NSR can maintain the exploration space of RL, but a larger exploration space is not always favorable, as in logical tasks.

demonstrate comparable effectiveness in strong alignment settings, PSR emerges as the more robust approach in challenging domains where models lack expertise.

## 6.2 DISCUSSION

The relationship between positive and negative samples in reinforcement learning is fundamentally connected to the exploration-exploitation trade-off, with entropy serving as a key mediator. To elucidate these dynamics in our experimental context, we examine how different sample types affect the exploration-exploitation balance through their impact on training entropy.

**Negative Signals Help Maintain Exploration.** Figure 4 plots token-level entropy throughout training. Consistent with Zhu et al. (2025), NSR slows entropy collapse, especially on mathematical tasks—suggesting that penalising only erroneous trajectories can preserve output diversity. However, the flatter entropy curve on logical tasks corresponds to poorer final accuracy.

## 7 MORE ANALYSIS

To further strengthen our hypothesis, we conduct several additional analyses:

**Our conclusion holds under broader settings.** Beyond Qwen and LLaMA and the two domains of mathematics and logic, we further validate our conclusions on code-generation tasks (Appendix G) and on Mistral (Appendix H), a model that performs poorly on both mathematics and logic. Across these broader settings, our alignment-based hypothesis continues to hold.

**KL change under standard RL as an auxiliary model–task alignment indicator.** In Appendix I, we show that, in addition to Pass@K, the magnitude of the KL divergence change induced by standard RL (before vs. after training) can serve as an auxiliary quantitative indicator of model–task alignment. We further verify that adding a KL penalty during training does not change our main conclusions.

**Pass@K strongly predicts RL gains.** Figure 10 is to directly visualize the relationship between our alignment proxy (Pass@K) and the gains from standard RL. Across different $K$ values (e.g., $K=32, 128, 512$) and across the RL phenomena we study, we consistently observe a positive trend: performance improvements grow as Pass@K increases.

## 8 CONCLUSION

This work reveals that *Model-Task Alignment* strength, measured by pass@k accuracy, serves as the fundamental determinant of when counterintuitive RL phenomena emerge in language model reasoning. We demonstrate that remarkable behaviors—including robustness to spurious rewards, one-shot training effectiveness, and negative-only signal sufficiency—manifest primarily when models already possess strong foundational capabilities in the target domain, functioning more as capability elicitation mechanisms rather than genuine learning drivers for unfamiliar tasks.

ACKNOWLEDGMENT

This project is partially supported by Hong Kong RGC ECS Grant 26218125, Hong Kong RGC CRF Grant C6003-24Y, and NSFC Grant 62306177.

REPRODUCIBILITY STATEMENT

To ensure full reproducibility of our results, we conduct all experiments using publicly available models and datasets. We provide complete implementation code, detailed hyperparameter configurations, and step-by-step reproduction instructions in the supplementary material.

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

## A    LLM Usage Statement

As non-native English speakers, we used LLMs solely to assist with grammatical correction and linguistic polishing of the manuscript. The LLM was not involved in any aspect of conceptual development, experimental design, data analysis, or interpretation of results. All scientific content, including hypotheses, methodology, figures, and conclusions, was generated independently by the authors. The use of the LLM was strictly limited to improving clarity and fluency of expression in English, ensuring that language barriers do not impede the accurate communication of our research contributions.

## B    Implementation Details

Following the setting described in Section 3, we train with different rewards for 300 steps on mathematical and logical reasoning tasks, respectively. The format reward is different from that of Shao et al. (2025), we use the same format as SynLogic:

<think>thinking process</think><answer>final answer</answer>

We set $\gamma = 0.5$ for the random reward. For incorrect rewards, we reward rollouts that produce incorrect answers during training for logic and code tasks. Apart from these changes, the definition of the reward functions remains consistent with spurious reward (Shao et al., 2025).

## C    More Pass@k Results

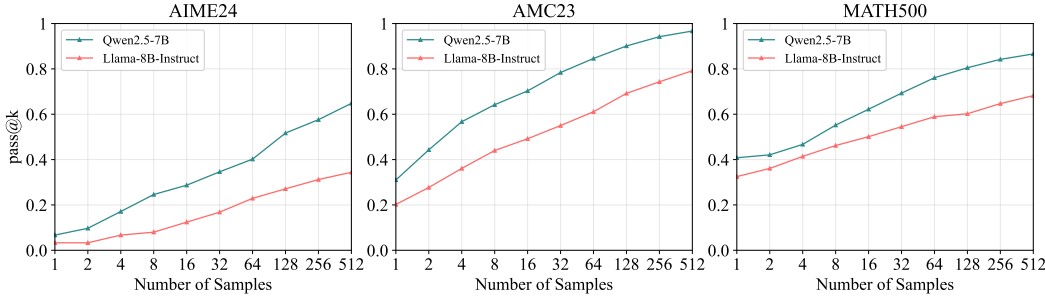

Figure 5: Pass@k for math tasks. Qwen demonstrates strong capabilities across all three mathematical evaluation datasets.

## D    Contamination Evaluation

### D.1    Implementation Details

Our contamination analysis follows a systematic prompt truncation methodology to evaluate potential data leakage across model-task combinations. Original prompts are truncated at varying ratios (0.4, 0.6, and 0.8) while preserving word boundaries, and models are asked to complete the remaining content using greedy decoding for deterministic outputs. We measure contamination using ROUGE-L scores between model completions and the actual remaining prompt content, where a perfect score of 1.0 indicates complete reconstruction and potential contamination. The evaluation pipeline employs distributed processing to handle complex mathematical expressions and prevent evaluation timeouts, with results aggregated across multiple rollouts to ensure statistical reliability.

### D.2    More Results

## E    Discussion about RQ1

**How Different Reward Signals Affect the Behavior of LLMs.**    Shao et al. (2025) observed that in mathematical tasks, employing ground truth rewards decreases the frequency of code usage in

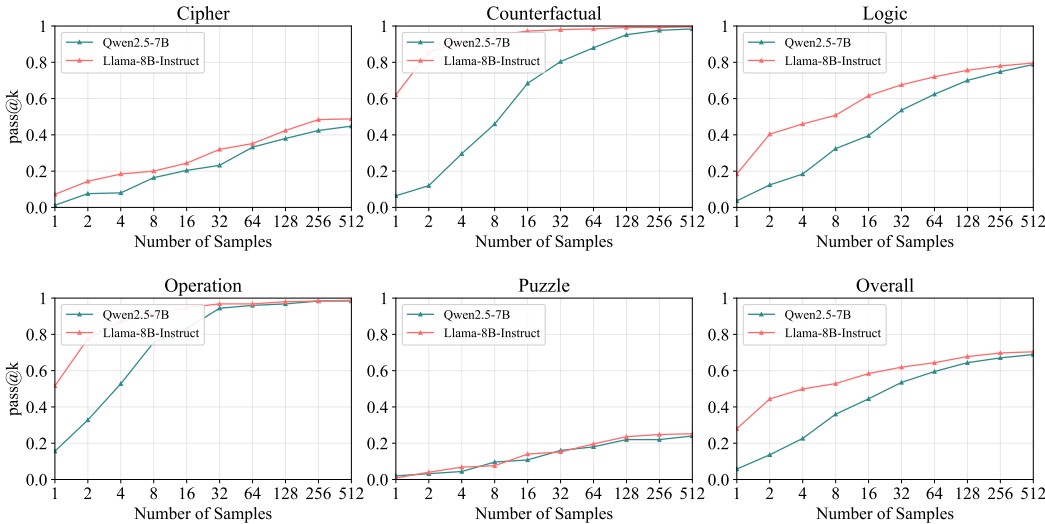

Figure 6: Pass@k for KOR-Bench. Both models demonstrate strong inherent reasoning capabilities in Operation and Counterfactual subtasks, but exhibit limited inherent logical reasoning abilities in Cipher, Puzzle and Logic.

| Task Type | Benchmark | Model | Portion=0.4 | | Portion=0.6 | | Portion=0.8 | |
|---|---|---|---|---|---|---|---|---|
| | | | ROUGE | EM | ROUGE | EM | ROUGE | EM |
| Math Tasks | AMC 23 | Qwen2.5-7B | 63.78 | 23.91 | 64.42 | 33.73 | 73.23 | 49.39 |
| | | Llama-3.1-8B | 27.18 | 0.00 | 30.64 | 0.00 | 44.54 | 4.81 |
| | MATH500 | Qwen2.5-7B | 50.36 | 8.20 | 60.98 | 21.20 | 66.42 | 40.20 |
| | | Llama-3.1-8B | 23.09 | 0.60 | 40.56 | 3.80 | 48.33 | 17.8 |
| | AIME24 | Qwen2.5-7B | 44.64 | 10.00 | 48.69 | 13.33 | 60.08 | 30.00 |
| | | Llama-3.1-8B | 26.08 | 0.00 | 30.80 | 0.00 | 50.50 | 13.33 |
| Logic Tasks | Puzzle | Qwen2.5-7B | 19.56 | 0.00 | 19.62 | 0.00 | 19.24 | 0.00 |
| | | Llama-3.1-8B | 18.27 | 0.00 | 17.31 | 0.00 | 15.85 | 0.00 |
| | Operation | Qwen2.5-7B | 21.37 | 0.00 | 24.25 | 0.00 | 20.18 | 0.00 |
| | | Llama-3.1-8B | 21.83 | 0.00 | 18.34 | 0.00 | 16.75 | 0.00 |
| | Counterfactual | Qwen2.5-7B | 18.88 | 0.00 | 19.96 | 0.00 | 18.66 | 0.00 |
| | | Llama-3.1-8B | 19.02 | 0.00 | 19.39 | 0.00 | 18.94 | 0.00 |
| | Logic | Qwen2.5-7B | 22.08 | 0.00 | 27.28 | 0.00 | 28.23 | 0.00 |
| | | Llama-3.1-8B | 21.38 | 0.00 | 28.37 | 0.00 | 28.42 | 0.00 |
| | Cipher | Qwen2.5-7B | 34.61 | 0.00 | 41.03 | 0.00 | 44.77 | 0.00 |
| | | Llama-3.1-8B | 29.59 | 0.00 | 36.95 | 0.00 | 42.93 | 0.00 |

Table 7: Extended Contamination Analysis across model-task combinations. **Red** indicates potential contamination with strong baseline performance; **Gray** indicates no contamination with weak baseline performance; **Green** indicates no contamination with strong baseline performance.

model responses. Their study also revealed that, in contrast to Qwen2.5-Math (Yang et al., 2024), the accuracy improvement of the Qwen2.5 Base model was primarily attributed to a shift from code-based reasoning to language-based reasoning. As shown in Table 8, we identify analogous trends in mathematical tasks. Specifically, for logic puzzles, the application of ground truth rewards similarly reduces the incidence of code in responses. However, other types of rewards, particularly

format and random rewards, do not demonstrate a significant impact on diminishing code usage frequency. We speculate that, throughout the RL training process, ground truth rewards can steer the model away from its old reasoning pattern ( i.e., producing reasoning responses with code ) and toward a more natural, language-based reasoning pattern.

| Reward Type | MATH500 | | SynLogic | |
| --- | --- | --- | --- | --- |
| | Before RL | After RL | Before RL | After RL |
| Correct | | **12.4** | | **21.7** |
| Random | 89.1 | 94.2 | 57.3 | 48.2 |
| Format | | 96.7 | | 50.7 |
| Incorrect | | 28.1 | | 28.3 |

Table 8: Code Usage Count of Qwen2.5-7B before and after RL training with different rewards.

As shown in Table 2, spurious rewards are effective only on the Operation and Counterfactual for the Llama model; consequently, we also report the frequency of code-based reasoning before and after training on these two tasks. As shown in Table 9, we observe that, both before and after RL training, Llama almost never invokes code during the reasoning process. We attribute the sporadic use of code (0.8) to the fact that some SynLogic tasks explicitly require outputs to be presented as code blocks. This indicates that Llama and Qwen exhibit distinct reasoning patterns even though they both benefit from noisy reward signals in these settings.

| Reward Type | Operation | | Counterfactual | |
| --- | --- | --- | --- | --- |
| | Before RL | After RL | Before RL | After RL |
| Correct | | 0.8 | | 0.0 |
| Random | 0.0 | 0.0 | 0.0 | 0.0 |
| Format | | 0.0 | | 0.0 |
| Incorrect | | 0.0 | | 0.0 |

Table 9: Code Usage Count of Llama-3.1-8B-Instruct before and after RL training on two tasks.

## F   MORE DISCUSSION ABOUT DIFFICULT EXAMPLE IN ONE-SHOT RL

During training with $l_{selected}$, apart from the rollout accuracy (reward) remaining consistently at 0, metrics such as entropy and response length also exhibit almost no changes. As shown in Figure 7, after 300 training steps, the model still maintains a large reinforcement learning exploration space.

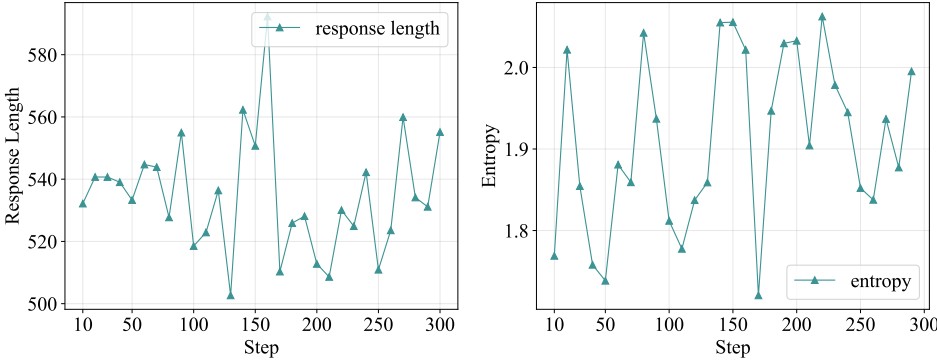

Figure 7: Training Dynamics of Qwen2.5-7B when trained with $l_{selected}$. Entropy and response length exhibit almost no changes.

## G  RESULTS ON CODE GENERATION TASKS

We additionally include code-generation tasks to further validate the generality of our findings. Specifically, we evaluate on HumanEval (Chen et al., 2021) and LiveCodeBench (2024.8–2025.1) (Jain et al., 2024). The live nature of LiveCodeBench ensures that data contamination is almost impossible. However, both models achieve relatively high Pass@1 scores on HumanEval. To assess whether these high scores may result from contamination, we conduct an analysis analogous to that in Table 1. The results are shown in Table 10. Similar to the math tasks, we observe a higher risk of data contamination in Qwen. Accordingly, we categorize Qwen/HumanEval as red and Llama/HumanEval as green. And we show the Pass@K curves for LiveCodeBench in Figure 8. Neither model exhibits sufficiently strong capability on this benchmark; as K increases, their performance does not rise as sharply as it does on tasks such as Operation (shown in Figure 6). Consequently, we categorize Qwen/LiveCode and Llama/LiveCode as gray.

| Model | Portion | ROUGE | EM | Model | Portion | ROUGE | EM |
|---|---|---|---|---|---|---|---|
| | 0.4 | 56.32 | 9.7 | | 0.4 | 23.42 | 0.0 |
| **Qwen2.5-7B** | 0.6 | 60.89 | 14.3 | **LLama3.1-8B** | 0.6 | 32.13 | 1.2 |
| | 0.8 | 66.81 | 36.7 | | 0.8 | 47.32 | 8.4 |

Table 10: Contamination Analysis on HumanEval.

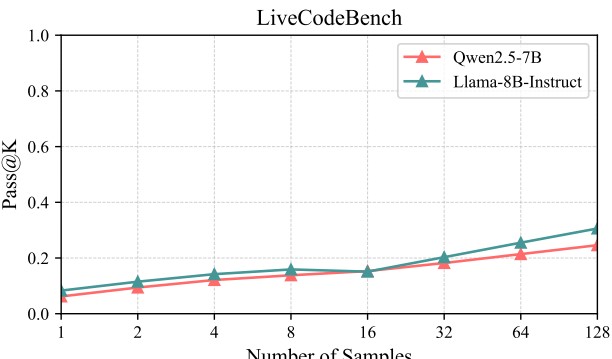

Figure 8: Pass@K curves for LiveCodeBench.

For training, we use code-r1-12k (Liu, 2025) as the training data. Importantly, to keep the setup consistent with the math and logic tasks, we assign a correctness reward of 1 only when the generated code passes all test cases; otherwise, the model receives no correctness reward at all. Because code outputs are difficult to aggregate via majority voting, we do not evaluate the Vote setting in this task. To reduce variance, we report avg@8, following the same protocol as SynLogic and AIME24.

| | HumanEval | LiveCodeBench | | HumanEval | LiveCodeBench |
|---|---|---|---|---|---|
| **Qwen2.5-7B** | 52.4 | 6.2 | **Llama3.1-8B** | 68.4 | 8.3 |
| Correct | $65.2_{+12.8}$ | $15.4_{+9.2}$ | Correct | $75.1_{+6.7}$ | $12.0_{+3.7}$ |
| Format | $58.5_{+6.1}$ | $5.2_{-1.0}$ | Format | $69.2_{+0.8}$ | $8.9_{+0.6}$ |
| Random | $56.1_{+3.7}$ | $5.0_{-1.2}$ | Random | $67.4_{-1.0}$ | $7.3_{-1.0}$ |
| Incorrect | $53.0_{+0.6}$ | $4.2_{-2.0}$ | Incorrect | $68.8_{+0.4}$ | $6.0_{-2.3}$ |
| EM | $58.1_{+5.7}$ | $6.2_{+0.0}$ | EM | $70.4_{+2.0}$ | $8.0_{-0.3}$ |
| 1-shot | $51.5_{-0.9}$ | $6.0_{-0.2}$ | 1-shot | $69.0_{+0.6}$ | $6.5_{-1.8}$ |
| NSR | $55.1_{+2.7}$ | $7.4_{+1.2}$ | NSR | $70.4_{+2.0}$ | $6.8_{-1.5}$ |

Table 11: Code-generation results on HumanEval and LiveCodeBench. The 1-shot samples are selected based on Wang et al. 2025's method.

The results of the two code evaluation tasks are shown in Table 11. From the table, we can see that under the standard RL setting, the model shows clear improvements on both HumanEval and LiveCodeBench. Under non-standard settings, only the red-category case—Qwen/HumanEval and the green-category case Llama/HumanEval—exhibit small gains, while no consistent improvement is observed in the weak-alignment setting. This is consistent with our observations on math and logic tasks.

## H  RESULTS OF A MORE WEAKLY ALIGNED MODEL: MISTRAL-7B-V0.1

We additionally include Mistral-7B-v0.1 (Jiang et al., 2023) in the weak-alignment setting. Its performance on math and logic tasks is worse than that of Llama-3.1-8B-Instruct, which allows us to provide more weakly aligned settings. Results are shown in Table 12. We can see that Mistral shows stable improvements on math and logic tasks only under the standard RL setting. This further reinforces the validity of our conclusions under the weak-alignment setting.

| | Math Tasks | | | Logic Tasks | | | | | | | |
| | | | | | | | KOR Benchmark | | | | |
| | AIME24 | MATH500 | AMC | SynLogic | BBH | BBEH | OP | CF | Puzzle | Logic | Cipher |
|---|---|---|---|---|---|---|---|---|---|---|---|
| | | | | **Mistral-7B-v0.1 (Weakly Aligned)** | | | | | | | |
| Base | 0.0 | 6.8 | 3.4 | 0.0 | 27.1 | 0.4 | 11.2 | 22.8 | 2.4 | 28.4 | 2.4 |
| | | | | RLVR (External Reward) | | | | | | | |
| GroundTruth | $3.3_{+3.3}$ | $17.6_{+10.8}$ | $12.5_{+9.1}$ | $14.6_{+14.6}$ | $38.9_{+11.8}$ | $3.2_{+2.8}$ | $26.8_{+15.6}$ | $43.4_{+20.6}$ | $9.8_{+7.4}$ | $46.2_{+17.8}$ | $8.8_{+6.4}$ |
| Format | $0.0_{+0.0}$ | $6.4_{-0.4}$ | $2.0_{-1.4}$ | $0.0_{+0.0}$ | $24.3_{-2.8}$ | $0.0_{-0.4}$ | $14.6_{+3.4}$ | $26.2_{+3.4}$ | $2.0_{-0.4}$ | $26.8_{-1.6}$ | $3.2_{+0.8}$ |
| Random | $0.0_{+0.0}$ | $6.4_{-0.4}$ | $3.2_{-0.2}$ | $0.4_{+0.4}$ | $21.2_{-5.9}$ | $0.4_{+0.0}$ | $9.2_{-2.0}$ | $19.4_{-3.4}$ | $2.0_{-0.4}$ | $22.4_{-6.0}$ | $2.0_{-0.4}$ |
| Incorrect | $0.0_{+0.0}$ | $5.6_{-1.2}$ | $1.0_{-2.4}$ | $0.0_{+0.0}$ | $13.4_{-13.7}$ | $0.0_{-0.4}$ | $9.0_{-2.2}$ | $19.2_{-3.6}$ | $1.8_{-0.6}$ | $21.0_{-7.4}$ | $1.8_{-0.6}$ |
| | | | | Self-Rewarded Reinforcement Learning | | | | | | | |
| Vote | $0.4_{+0.4}$ | $5.8_{-1.0}$ | $3.2_{-0.2}$ | $0.0_{+0.0}$ | $14.2_{-12.9}$ | $0.4_{+0.0}$ | $12.8_{+1.6}$ | $28.6_{+5.8}$ | $1.2_{-1.2}$ | $18.8_{-9.6}$ | $3.6_{+1.2}$ |
| EM | $0.0_{+0.0}$ | $7.8_{+1.0}$ | $2.4_{-1.0}$ | $0.8_{+0.8}$ | $23.4_{-3.7}$ | $0.4_{+0.0}$ | $10.4_{-0.8}$ | $19.2_{-3.6}$ | $1.4_{-1.0}$ | $24.4_{-4.0}$ | $2.0_{-0.4}$ |
| | | | | Few-shot Reinforcement Learning | | | | | | | |
| 1-shot (selected) | $0.4_{+0.4}$ | $8.4_{+1.6}$ | $2.8_{-0.6}$ | $0.0_{+0.0}$ | $22.4_{-4.7}$ | $0.0_{-0.4}$ | $8.8_{-2.4}$ | $19.8_{-3.0}$ | $2.4_{+0.0}$ | $29.6_{+1.2}$ | $2.0_{-0.4}$ |
| | | | | Negative Sampling Reinforcement Learning | | | | | | | |
| NSR | $0.8_{+0.8}$ | $6.4_{-0.4}$ | $3.0_{-0.4}$ | $0.0_{+0.0}$ | $24.2_{-2.9}$ | $0.4_{+0.0}$ | $12.4_{+1.2}$ | $23.2_{+0.4}$ | $2.8_{+0.4}$ | $24.0_{-4.4}$ | $2.0_{-0.4}$ |

Table 12: Results of Mistral-7B-v0.1.

### H.1  DISCUSSION

In Tables 2 and 6, although Llama–Math is a weakly aligned model–task pair, both self-rewarded methods and NSR consistently lead to performance improvements. Here we provide an explanation for this phenomenon: Although we categorize Llama–Math as a weakly aligned model–task pair, its alignment is still noticeably stronger than that of Llama–Logic. We see two reasons for this:

1. As shown in Table 1, although we do not observe the severe contamination found in Qwen, the Llama–Math combination exhibits a higher risk of data contamination compared with Llama–Logic (EM = 0)
2. As shown in Figure 5, while the model's performance does not increase rapidly as K grows, its curve is still significantly sharper than that of tasks such as Puzzle in Figure 6

That being said, model–task alignment strength may be more continuous rather than falling into just three discrete categories. When the alignment is slightly stronger, as in the case of LLama on math tasks, some RL methods may begin to take effect. The results on Mistral further support our conclusion: due to its weaker alignment, neither NSR nor the self-reward method achieves consistent performance gains on the math tasks. Identifying such an "emergence boundary" in terms of model–task alignment strength is an interesting direction for future work.

# I  KL DIVERGENCE BETWEEN THE INITIAL AND TRAINED POLICIES

During training, consistent with DAPO (Yu et al., 2025), we removed the KL regularization term from the loss function. This enables us to fairly compare the impact of different training methods on the output distribution under varying levels of alignment.

We explore it in three representative settings: Qwen on MATH500 (red), Qwen on Operation (green), and Qwen on SynLogic (gray). Specifically, we use trained policies to generate trajectories via greedy decoding. Then, we feed these trajectories into the untrained reference model to obtain the log-probabilities for each token. We follow the approach used in DeepSeek-R1 (Guo et al., 2025) for computing KL divergence:

$$\mathrm{KL}(\pi_\theta || \pi_{ref}) = \frac{\pi_{ref}(o_i|q, o_{<i})}{\pi_\theta(o_i|q, o_{<i})} - \log \frac{\pi_{ref}(o_i|q, o_{<i})}{\pi_\theta(o_i|q, o_{<i})} - 1 \tag{1}$$

and compute the average KL over all tokens. We present the results in the Figure 9. From the results, we can observe that:

- In the standard RL setting, weaker model–task alignment leads to a larger divergence between the pre- and post-training output distributions (the gray is substantially higher than the green and red ones).
- In the weak-alignment setting (gray), only standard RL and PSR can drive the model away from the reference model and achieve substantial performance improvements (+41.1 and +23.3 on SynLogic).
- In the strong-alignment setting (green and red), neither the reward choices nor the sampling methods substantially amplify the divergence between the pre- and post-training output distributions.

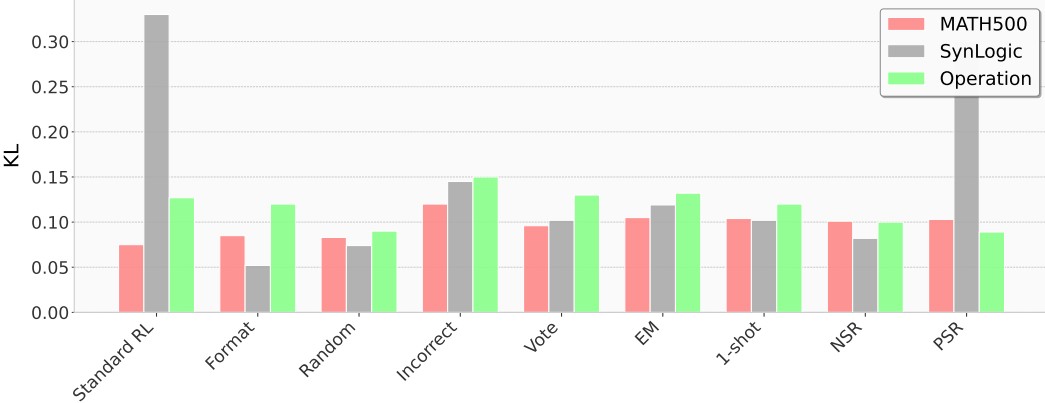

Figure 9: KL divergence between the initial and trained policies under different settings.

Based on these results, we hypothesize that in the strong-alignment setting, large deviations from the initial policy are unnecessary; small updates are sufficient to improve performance. This also helps explain why spurious-reward methods can still be effective, as the initial policy is already close to the correct solution. In contrast, in the weak-alignment setting, merely oscillating around the initial policy does not yield meaningful gains—substantial improvement requires moving further away under the guidance of correct rewards to form a stable and effective reasoning pattern.

## I.1  POTENTIAL CHANGES INTRODUCED BY ADDING KL REGULARIZATION

A natural question is whether introducing KL regularization during training would affect our conclusions. Considering that the difference between the pre- and post-training output distributions is small in the strong-alignment setting, we believe that introducing KL regularization does not affect the overall trend of performance changes, although it may influence convergence speed (Yu et al., 2025). To assess the impact under the weak-alignment setting, particularly for standard RL and PSR, we set the KL penalty coefficient to 0.001 and retrain Qwen on the logic tasks. The experimental

results are shown in Table 13. Introducing KL regularization leads to a slight drop in performance. However, the overall trend remains the same as in the setting without regularization, indicating that the presence or absence of KL regularization does not affect our conclusions.

| | SynLogic | BBH | BBEH |
|---|---|---|---|
| Pretrain | 1.5 | 45.2 | 1.2 |
| Standard | $42.6_{+41.1}$ | $62.7_{+17.5}$ | $6.8_{+5.6}$ |
| Standard + KL | $33.6_{+32.1}$ | $52.1_{+6.9}$ | $6.8_{+5.6}$ |
| PSR | $24.8_{+23.3}$ | $57.1_{+11.9}$ | $4.3_{+3.1}$ |
| PSR + KL | $17.3_{+15.8}$ | $49.1_{+3.9}$ | $3.8_{+2.6}$ |

Table 13: Performance of different training methods with and without KL regularization under the weak-alignment setting.

## I.2 DISCUSSION

From Figure 9, we believe that the divergence between the pre- and post-training output distributions in a successful standard RL run (without collapse) can serve as a complementary indicator of model–task alignment strength. This is also consistent with our intuition: in the strong-alignment setting, the model does not need to extensively explore regions far from the initial policy to achieve performance gains. In contrast, in the weak-alignment setting, a successful training process must sufficiently explore regions farther away from the initial policy in order to discover an effective reasoning pattern. Moreover, the consistency between KL divergence and Pass@K further reinforces that Pass@K can be viewed as a reliable proxy for model–task alignment.

## J CORRELATION BETWEEN PASS@K AND RL GAINS

In Figure 10, we present the relationship between RL gains and Pass@K. Each data point in the figure is derived from the results reported in Tables 2, 3, 5, and 6. And each point represents a task, with points in the first column averaged over different reward signals. We can find that across different K values (32, 128, 512) and across the various RL phenomena we study, we consistently observe that performance improvements grow as Pass@K increases. This further strengthens the connection between model–task alignment strength and performance gains.

## K FEW-SHOT RL EXAMPLE DETAILS

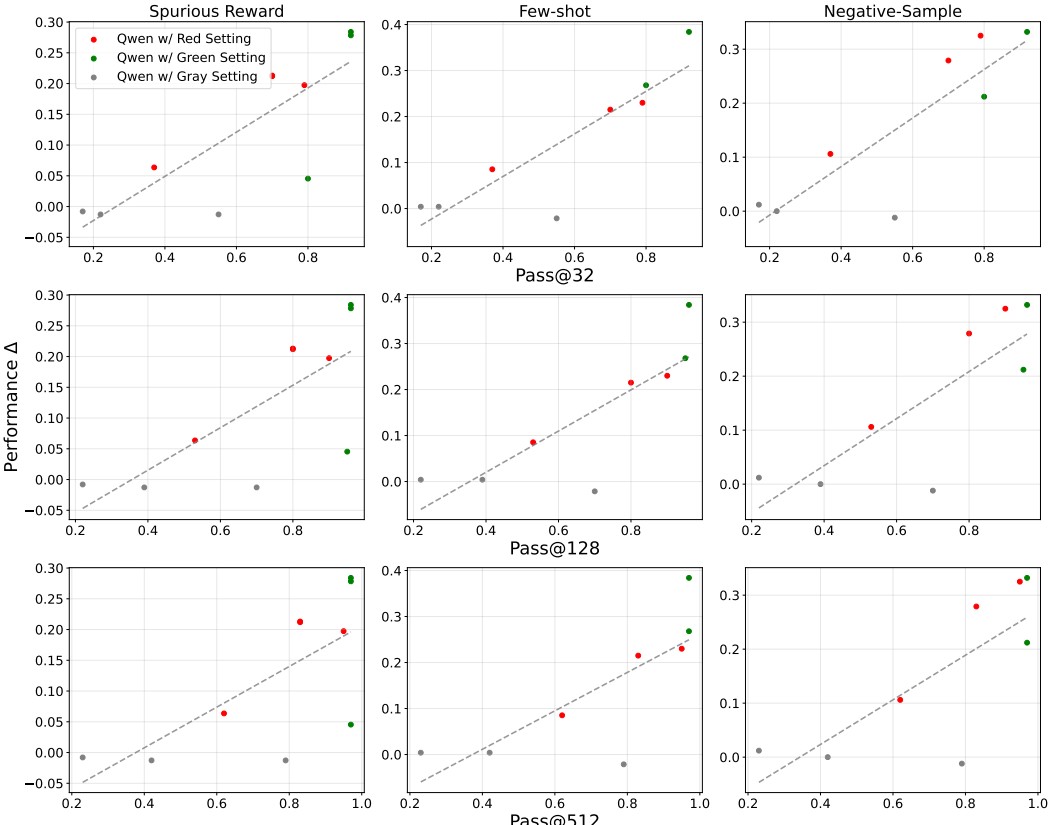

Figure 10: Pass@K versus performance gains across tasks and RL settings.

**Details of example $m_{selected}$**

How many positive divisors do 9240 and 13860 have in common?

**Details of example $m_{random}$**

The angles of quadrilateral $PQRS$ satisfy $\angle P = 3\angle Q = 4\angle R = 6\angle S$. What is the degree measure of $\angle P$?

**Details of example $m'_{random}$**

```
Given a finite sequence S   =   (a₁, a₂, ..., aₙ) of n real numbers,
```
let $A(S)$ be the sequence $\left(\frac{a_1+a_2}{2}, \frac{a_2+a_3}{2}, \ldots, \frac{a_{n-1}+a_n}{2}\right)$ of $n - 1$
real numbers. Define $A^1(S) = A(S)$ and, for each integer $m$,
$2 \leq m \leq n-1$, define $A^m(S) = A(A^{m-1}(S))$. Suppose $x > 0$, and let
$S = (1, x, x^2, \ldots, x^{100})$. If $A^{100}(S) = \left(\frac{1}{2^{50}}\right)$, then what is $x$? **AND** If
$x, 2x+2, 3x+3, \ldots$ are in geometric progression, the fourth term
is:

**Details of example $l_{selected}$**

```
Here's a mathematical expression:  ?-?+(6%5)*2-?+?/?/?/4/2
= 2.  The digits on the left side of the equation have been
replaced with question marks.  Each question mark corresponds
to a digit between 0 and 9.  You need to try replacing
the question marks with the correct digits to restore the
expression.Please put the complete expression with the filled
- in digits between [[ and ]] at the end of your response,
with no other content, like this:  [[2 + 4 * 3 - 4 = 10]]
```

**Details of example $l_{random}$**

```
Solve this cryptarithm:  RRYUU + UYR + U = RYUUU (where
RRYUU is a 5-digit number, UYR is a 3-digit number, U is a
1-digit number, and RYUUU is a 5-digit number).  Each letter
represents a unique digit.  Find the digit substitution that
makes the equation true.
```

**Details of example $l'_{random}$**

```
In this Number Wall puzzle, add walls (marked as 'A') to
divide the grid into islands.  Each island must contain
exactly one number, and its size must equal that number.
Grid:
+---+---+---+
| X | 3 | X |
+---+---+---+
| X | X | X |
+---+---+---+
| X | X | X |
+---+---+---+
Rules:
- Each island must contain exactly one number.
- The total number of cells in an island (including the
number cell) must equal the value of that number.
- All cells within an island must be connected horizontally
or vertically.
- Walls (marked as 'A') cannot form 2×2 or larger continuous
rectangles.
- All islands must be separated by walls.
```
**AND**
```
In the cryptarithm:  MMII + MIXIMM = MMXIIX, each letter
stands for a different digit (MMII is 4 digits, MIXIMM is 6
digits, and MMXIIX is 6 digits).  Determine what each letter
represents to make the equation true.
```

**Details of example $l_{simple}$**

```
In this word sorting challenge, you need to rearrange
words in increasing based on a modified alphabet
where l,z and a are the first letters.  Words to sort:
yachted,coelomic,harateen.  Write your final answer inside:
\boxed,like this:  \boxedword1,word2,word3.
```

**Details of example $l_{mid}$**

```
You are an expert proficient in Dyck language, where you must
complete all types of unclosed brackets (e.g., [], , <>) in
language sequences.  You need to analyze the steps of bracket
pairing according to Dyck language rules.  Given an initial
Dyck language sequence and steps for deriving the closed
bracket sequence (presented in a thinking process format),
your task is to identify locations with incorrect reasoning
in the Dyck language, and there may be multiple errors.  This
could be forgetting to close a bracket, using the wrong
closing bracket, or incorrectly copying a subsequence of
closing brackets in the next step.  Task:  Check the sequence
to ensure brackets are properly closed.  Input:  [[(){}]]{}
Thought 1:  We should process the input one by one and track
the stack configuration.
Thought 2:  Stack:  Empty
Thought 3:  [ ; Stack:  Empty
Thought 4:  [ ; Stack:  [[
Thought 5:  ( ; Stack:  [[(
Thought 6:  )  ; Stack:  [[
Thought 7:  { ; Stack:  [[{
Thought 8:  } ; Stack:  [[
Thought 9:  ]  ; Stack:  [
Thought 10:  ]  ; Stack:  Empty
Thought 11:  { ; Stack:  {
Thought 12:  } ; Stack:  Empty
Thought 13:  Now, we have reached the end.  The final stack
is empty.
Question:  Are there any reasoning errors in this sequence?
```

