# OpenReview forum: "Mirage or Method? How Model–Task Alignment Induces Divergent RL Conclusions"
_ICLR.cc/2026/Conference — ICLR 2026 Poster_

### Official Review · Reviewer_QY7Y · 2025-10-28

**Soundness:** 3
**Presentation:** 4
**Contribution:** 2
**Rating:** 6
**Confidence:** 4

**Summary:**

This papers presents an empirical study on some reported phenomenas in RL training such as: 1) training with random erroneous rewards (random reward), 2)  self rewarding LLMs via Test time RL i,e using the majority vote answer from a rollout as a true one , 3)  one shot , i,e trainig with a single selected example 4) GRPO training with only positive or negative sampling. Some recent work Wu et al 2025 explained some of these phenomenas as due to the contamination of the training set of some of LLMs with the test set. This paper probes another hypothesis that the contamination is not the only reason behind such phenomenas but the alignment between the task and the LLM at hand. The alignment between task and LLM is measure with pass@k of the LLM on the test set of this task. Authors focus on Qwen and Llama family of models and divide task and models to three areas: strong alignment and strong contamination (red), strong alignment and weak contamination (green) , and weak alignment and no contamination (gray). Throughout extensive experimentation authors find that pass@k on a task is a predictive of the performance of the model with standard RL, but also its robustness to the phenomenas listed above. A high pass@k wether contaminated or not pretaining translates into a lift in performance for standard RL and the variants above (TTRL, one shot RL, random reward, positive , negative sampling), and low pass@k leads to a bad performance across the board.

**Strengths:**

The paper is very well written and a very nice and easy read with all experiments well articulated and well presented, I personally enjoyed reading the paper.

The fact that pass@k of the reference model is a predictive of the performance of standard RL is not surprising and it has been even shown theoretically in multiple works showing that best of n is an approximation of the RL optimal policy see for example best of n through the smoothing lengths or in Asymptotics of Language Model Alignment or in  information theoretic gaurantees of alignment in LLMs. It is interesting to develop similar results to robustify these results to data and reward perturations as presented in this paper.

**Weaknesses:**

One interesting point the paper is not studying in the analysis is to follow the KL(pi|| pi_{ref}) when using standard RL versus the non standard RL (TTRL, one shot RL, PSR, NSR).

It would be interesting to compare the E_{test set }KL(pi|| pi_{ref}) , to see how the perturbations of the rewards/ sampling impacts the deviation from the reference models, in the green, red and gray area. Note that using DAPO we don't have the regularization to pi_{ref} so this is a fair study.

**Questions:**

* what is the group size you used in DAPO?
* do you think the conclusions will change if we add the KL regularization to pi_{ref} in the training of these RL ?
* can you study the KL of those perturbed RL to see how the perturbation of the reward impacts the KL emprically ?

---

> ### Author Response · Authors · 2025-11-24
> **Response to Reviewer QY7Y**
>
> Thank you for your time and encouraging review! We address your questions below:
>
> > **Can you study the KL of those perturbed RL to see how the perturbation of the reward impacts** **the** **KL** **emprically** **?**
>
> Thank you for the advice. We agree that comparing KL is a very interesting direction. Therefore, we explore it in three representative settings: Qwen on MATH500 (red), Qwen on Operation (green), and Qwen on SynLogic (gray). Specifically, we use trained policies to generate trajectories via greedy decoding. Then, we feed these trajectories into the untrained reference model to obtain the log-probabilities for each token. We follow the approach used in DeepSeek-R1 [1] for computing the KL divergence:
>
> $$\text{KL}(\pi_\theta||\pi_{ref})=\frac{\pi_{ref}(o_i|q,o_{<i})}{\pi_{\theta}(o_i|q,o_{<i})}-\log \frac{\pi_{ref}(o_i|q,o_{<i})}{\pi_{\theta}(o_i|q,o_{<i})} - 1$$
>
> and compute the average KL over all tokens. The results are shown in the following table.
>
> |                   | Standard RL | Format | Random | Incorrect | Vote  | EM    |
> | ----------------- | ----------- | ------ | ------ | --------- | ----- | ----- |
> | MATH500 （red)    | 0.075       | 0.085  | 0.083  | 0.120     | 0.096 | 0.105 |
> | SynLogic (gray)   | **0.33**    | 0.052  | 0.074  | 0.145     | 0.102 | 0.119 |
> | Operation (green) | 0.127       | 0.120  | 0.090  | 0.150     | 0.130 | 0.132 |
>
> We also present the results in the appendix of the revised version of the paper (Figure 9). From the results, we can observe that:
>
> - In the standard RL setting with correct rewards, weaker model–task alignment leads to a large divergence between the pre- and post-training output distributions, while the KL divergence is small for strong model-task alignment cases. This is intuitive, as when the initial model is weak at the task, RL training with correct rewards yields larger distribution changes. In contrast, if the initial model is already strong on the task, only a small change is sufficient to achieve a good training reward.
> - In the settings with perturbed reward, the KL divergence is generally small regardless of the model-task alignment status. We hypothesize that this is because the perturbed reward is noisy and confusing to the models, failing to provide a strong signal to guide models towards a direction stably.
>
> > **Do you think the conclusions will change if we add the KL regularization to pi_{ref} in the training of these RL ?**
>
> We analyze how the KL regularization loss under different training settings before and after fine-tuning. Considering that this KL divergence is small in the strong model-task alignment setting, as indicated in the last response, we believe that introducing KL regularization does not affect the overall trend of performance changes in such cases, although it may influence convergence speed. Thus, we conduct experiments to assess the impact under the weak-alignment setting due to limited time during rebuttal, particularly for standard RL and PSR. Specifically, we set the KL penalty coefficient to 0.001 and retrained Qwen on the logic tasks. The experimental results are shown in the table below (Table 13 in the revised paper):
>
> |               | SynLogic     | BBH          | BBEH       |
> | ------------- | ------------ | ------------ | ---------- |
> | Pretriain     | 1.5          | 45.2         | 1.2        |
> | Standard      | 42.6 (+41.1) | 62.7 (+17.5) | 6.8 (+5.6) |
> | Standard + KL | 33.6 (+32.1) | 52.1 (+6.9)  | 6.8 (+5.6) |
> | PSR           | 24.8 (+23.3) | 57.1 (+11.9) | 4.3 (+3.1) |
> | PSR + KL      | 17.3 (+15.8) | 49.1 (+3.9)  | 3.8 (+2.6) |
>
> Introducing KL regularization leads to a slight drop in performance. However, the overall trend remains the same as in the setting without regularization. We will add experiments for the strong alignment setting as well to complete this analysis, while we believe the KL loss will have a smaller effect in that case, since the KL divergence value is small anyway there.
>
> > **What is the group size you used in DAPO?**
>
> We use a group size of 16. In the revised version of the paper, we have clarified the experimental details to avoid potential confusion for readers.
>
>
> [1] DeepSeek-R1: Incentivizing Reasoning Capability in LLMs via Reinforcement Learning

---

### Official Review · Reviewer_nBRb · 2025-10-30

**Soundness:** 3
**Presentation:** 3
**Contribution:** 3
**Rating:** 8
**Confidence:** 2

**Summary:**

This paper examines various hypothesis and claims about the performance of RL methods for language models. In particular, they investigate numerous phenomena around RL -- training with noisy rewards, training on single examples etc. and show how their performance is strongly correlated with "model-task alignment".

**Strengths:**

* the paper is well organized and generally easy to follow. the purpose of each section is clearly defined.
* the experiments cover a number of different hypotheses for why certain phenomena appear
* The authors consider multiple model types.
* The insights provided by the paper are generally useful
* using pass@k for model-task alignemnt is an interesting idea, but digging more into the realm of RL I think this might really be an indicator of how much exploration is necessary for RL to work.

**Weaknesses:**

The paper claims that "model-task alignment" measured by pass@k is predictive of whether or not several of these RL-based phenomena occur. However, the authors only use a handful of benchmarks and bin the "model-task alignment" into either strong or weak. It would be interesting to see if the authors could plot the pass@k (the proposed proxy metric) versus the increase in performance to actually see the correlation. From the results pass@k does seem predictive, but it's hard to know for sure without a ton of information. Doing more tasks for all methods would be excessive, but it could be interesting to see at least one representative.   For example, one could consider a coding domain.

**Questions:**

not any at this time.

---

> ### Author Response · Authors · 2025-11-24
> **Response to Reviewer nBRb**
>
> Thank you for your time and encouraging review! We address your questions below:
>
> > **It would be interesting to see if the authors could plot the pass@k (the proposed proxy metric) versus the increase in performance to actually see the correlation.**
>
> Thank you for the suggestion! In response, we have added scatter plots in the appendix (Figure 10 in the revised paper) showing Pass@K versus performance gains. Across different K values (32, 128, 512) and across the various RL phenomena we study, we consistently observe that performance improvements grow as Pass@K increases.

---

### Official Review · Reviewer_FtyV · 2025-10-31

**Soundness:** 3
**Presentation:** 3
**Contribution:** 3
**Rating:** 6
**Confidence:** 4

**Summary:**

The paper identifies Model-Task Alignment, measured by pass@k accuracy on the evaluated task, as a key factor that explains the counterintuitive phenomena that have been reported in LLMs but not observed in traditional RL settings. These phenomena include single training example matching the performance on the entire dataset, the reward signal not needing to be very accurate, training only with negative samples matching the performance of sophisticated reward-based methods, efficacy of test-time RL, etc. The paper demonstrates across multiple model-task combinations, spanning two language models (Llama and Qwen), and multiple mathematical and logical reasoning domains, that strong model-task alignment explains the counterintuitive phenomena observed in LLMs but not in traditional RL. In settings with weak model-task alignment, standard RL approaches work while the counterintuitive techniques fail

**Strengths:**

The paper critically evaluates existing claims in RL for LLMs, such as the effectiveness of spurious rewards, one-shot training, and negative-only signal sufficiency, and identifies Model-Task Alignment as a key factor determining if these claims hold true.
Additionally, the paper provides a comprehensive and systematic examination of these counterintuitive reinforcement learning (RL) phenomena in large language models (LLMs), supported by rigorous experimental validation across different model architectures and task domains. Furthermore, they show that dataset contamination is not the underlying cause, as was proposed in earlier works, by demarcating model-task combinations into those potentially contaminated and those not potentially contaminated in their experiments. Moreover, the paper is well-written and easy to follow, with dedicated sections for each of the reported counterintuitive phenomena.

**Weaknesses:**

The experiments primarily focus on Qwen and Llama models (of similar parameter scale, 8b), which may limit the generalizability of the results to other LLMs. Further experiments encompassing more LLMs and model parameter scales would help to understand if the results are more general. Additionally, while the Model-Task Alignment hypothesis is well-supported, the paper could explore other factors that might also contribute to the observed phenomena, such as the role of pretraining data, model sizes, or architectural differences.

Additionally, the paper does not explicitly mention if hyperparameter search was performed for each model-task combination and the counterintuitive RL phenomenon being explored (eg: negative samples, spurious rewards, single example training, test-time RL etc.).  Different combinations might need different hyperparameters to elicit their best possible performance

**Questions:**

1) Do you have the model-task alignment results for other LLMs (eg: mistral, deepseek, gemma) or different model sizes?

1) Was hyperparameter tuning performed for each model-task combination and the counterintuitive RL phenomenon being explored?

2) Why does self-rewarded RL seem to work for the Llama-math combination in Table 2, which has no contamination with weak model-task alignment? Any factors that may explain this?

3) Similar to the previous question, Negative Sampling Reinforcement (NSR) also seems to work for the Llama-math combination in Table-5. What explains these observations?

---

> ### Author Response · Authors · 2025-11-24
> **Response to Reviewer FtyV (1/2)**
>
> Thank you for your time and encouraging review! We address your questions below:
>
> > **Why does self-rewarded RL seem to work for the Llama-math combination in Table 2, which has no contamination with weak model-task alignment? Any factors that may explain this?**
> >
> > **Similar to the previous question, Negative Sampling Reinforcement (NSR) also seems to work for the Llama-math combination in Table-5. What explains these observations?**
>
> Although we categorize Llama–Math as a weakly aligned model–task pair, its alignment is still noticeably stronger than that of Llama–Logic. We see two reasons for this.
>
> 1. As shown in Table 1, although we do not observe severe contamination like that in Qwen, the Llama–Math combination exhibits a higher risk of data contamination compared with Llama–Logic (EM = 0).
> 2. As shown in Figure 5, while the model’s performance does not increase rapidly as K grows, its curve is still significantly sharper than that of tasks such as Puzzle in Figure 6.
>
> That being said, the model-task alignment strength may be more continuous than just three discrete categories. When the model-task alignment is slightly stronger, like LLama-math, certain RL methods may start to work there. Identifying such an "emergence boundary" in terms of model-task alignment strength is an interesting future direction to explore.
>
> > **Do you have the model-task alignment results for other LLMs (eg: mistral, deepseek, gemma) or different model sizes?**
>
> To answer this question, we select Mistral-7B-v0.1, which exhibits low Pass@K on both math and logical reasoning tasks, indicating weaker alignment. The results are shown in the table below (Table 12 in the revised paper):
>
> |                     | MATH         | AIME24     | AMC23       | SynLogic     | BBH          | BBEH       | OP           | CF           | Puzzle     | Logic        | Cipher     |
> | ------------------- | ------------ | ---------- | ----------- | ------------ | ------------ | ---------- | ------------ | ------------ | ---------- | ------------ | ---------- |
> | **Mistral-7B-v0.1** | **6.8**      | **0.0**    | **3.4**     | **0.0**      | **27.1**     | **0.4**    | **11.2**     | **22.8**     | **2.4**    | **28.4**     | **2.4**    |
> | GroundTruth         | 17.6 (+10.8) | 3.3 (+3.3) | 12.5 (+9.1) | 14.6 (+14.6) | 38.9 (+11.8) | 3.2(+2.8)  | 26.8 (+15.6) | 43.4 (+20.6) | 9.8 (+7.4) | 46.2 (+17.8) | 8.8 (+6.4) |
> | Format              | 6.4 (-0.4)   | 0.0 (0.0)  | 2.0 (-1.4)  | 0.0 (0.0)    | 24.3 (-2.8)  | 0.0 (-0.4) | 14.6 (+3.4)  | 26.2 (+3.4)  | 2.0 (-0.4) | 26.8 (-1.6)  | 3.2 (+0.8) |
> | Random              | 6.4 (-0.4)   | 0.0 (0.0)  | 3.2 (-0.2)  | 0.4 (+0.4)   | 21.2 (-5.9)  | 0.4 (0.0)  | 9.2(-2.0)    | 19.4 (-3.4)  | 2.0 (-0.4) | 22.4 (-6.0)  | 2.0 (-0.4) |
> | Incorrect           | 5.6 (-1.2)   | 0.0 (0.0)  | 1.0 (-2.4)  | 0.0 (0.0)    | 13.4 (-13.7) | 0.0 (-0.4) | 9.0 (-2.2)   | 19.2 (-3.6)  | 1.8 (-0.6) | 21.0 (-7.4)  | 1.8 (-0.6) |
> | Vote                | 5.8 (-1.0)   | 0.4 (+0.4) | 3.2 (-0.2)  | 0.0 (0.0)    | 14.2 (-12.9) | 0.4 (0.0)  | 12.8(+1.6)   | 28.6 (+5.8)  | 1.2 (-1.2) | 18.8 (-9.6)  | 3.6 (+1.2) |
> | EM                  | 7.8 (+1.0）  | 0.0 (0.0)  | 2.4 (-1.0)  | 0.8 (+0.8)   | 23.4 (-3.7)  | 0.4 (0.0)  | 10.4 (-0.8)  | 19.2 (-3.6)  | 1.4 (-1.0) | 24.4 (-4.0)  | 2.0 (-0.4) |
> | 1-shot (selected)   | 8.4 （+1.6） | 0.4 (+0.4) | 2.8 (-0.6)  | 0.0 (0.0)    | 22.4 (-4.7)  | 0.0 (-0.4) | 8.8 (-2.4)   | 19.8 (-3.0)  | 2.4 (0.0)  | 29.6 (+1.2)  | 2.0 (-0.4) |
> | NSR                 | 6.4 （-0.4） | 0.8 (+0.8) | 3.0 (-0.4)  | 0.0 (0.0)    | 24.2 (-2.9)  | 0.4 (0.0)  | 12.4 (+1.2)  | 23.2 (+0.4)  | 2.8 (+0.4) | 24.0 (-4.4)  | 2.0 (-0.4) |
>
> It can be seen that when the model is less aligned with the task, methods such as NSR and self-reward no longer produce consistent improvements on math tasks. Moreover, because the model and task are poorly aligned, only the standard RL setting is able to deliver consistent performance improvements.
>
> Due to limitations in time and computational resources, we are unable to provide results for larger-scale models. However, we believe the three models and tasks we selected are highly representative, as they collectively cover the red, green, and gray task categories.

---

> ### Author Response · Authors · 2025-11-24
> **Response to Reviewer FtyV (2/2)**
>
> > **Was hyperparameter tuning performed for each model-task combination and the counterintuitive RL phenomenon being explored?**
>
> We did not perform  hyperparameter tuning for different model-task experiments. Instead, we deliberately keep the learning rate, batch size, and other key hyperparameters identical across all experiments and align them with those used in SynLogic [1]. Our goal is to minimize confounding factors and ensure that differences in outcomes primarily reflect differences in Model–Task Alignment rather than tuning effort. While this means that certain model–task combinations may not achieve their absolute best performance, the qualitative trends (whether the performance improves as substantially as with standard RL) remain stable under consistent hyperparameters.
>
> [1] SynLogic: Synthesizing Verifiable Reasoning Data at Scale for Learning Logical Reasoning and Beyond

---

### Official Review · Reviewer_DMgH · 2025-11-02

**Soundness:** 2
**Presentation:** 2
**Contribution:** 2
**Rating:** 4
**Confidence:** 4

**Summary:**

This paper argues that many of the recent, slightly “too good to be true” RL-for-LLMs findings (spurious/noisy rewards still work, one-shot RL, test-time RL, negative-only signals) only work when the base model is already well aligned with the task, and that this alignment can be read off from pass@k on that task. When alignment is weak, they say, those tricks collapse, while plain RLVR/DAPO still helps.

**Strengths:**

This works has studied different claims in RL literature for LLM to show their effectiveness.

**Weaknesses:**

1. The experimental evidence is drawn almost entirely from Qwen2.5-7B and Llama-3.1-8B on math and KOR-style reasoning tasks (AIME24, MATH500, AMC, Operation, Counterfactual, Puzzle, Cipher). The conclusion that model–task alignment is a *fundamental* determinant therefore generalises beyond the evaluated model/task scope.

2. The paper equates model–task alignment with pass@k, but no prior work is cited to justify this choice of proxy. It would strengthen the argument to motivate pass@k more formally or to report an additional alignment signal (e.g. a divergence-based measure between base and trained models) as a complementary view.

3. The description in Appendix D.3 of “incorrect” and “format” rewards is underspecified: the exact criteria for each reward type and the RL setup used in that subsection should be made explicit, and the authors could briefly justify why they chose this particular RL recipe over newer GRPO-style variants (e.g., Dr.GRPO).

4. The set of weakly aligned models is fairly narrow (mainly Llama-3.1-8B in low-pass@k regimes). Including another family with weaker out-of-the-box performance on these tasks (e.g. Gemma 3) would make the alignment–effectiveness pattern more convincing.

5. The paper should state earlier and more prominently that DAPO is the default RL method used in the experiments, so readers do not have to infer it from later sections/appendices.

**Questions:**

see weaknesses

---

> ### Author Response · Authors · 2025-11-24
> **Response to Reviewer DMgH (1/3)**
>
> Thank you for your time and helpful comments! We address your concerns below:
>
> > **The experimental evidence is drawn almost entirely from Qwen2.5-7B and Llama-3.1-8B on math and KOR-style reasoning tasks. The** **conclusion** **that model–task alignment is a** ***fundamental*** **determinant** **therefore generalises beyond the evaluated model/task scope.**
>
> Thanks for the suggestion, we additionally include code-generation tasks to further validate the generality of our findings. Specifically, we evaluate on HumanEval [1] and LiveCodeBench (2024.8–2025.1) [2]. The live nature of LiveCodeBench ensures that data contamination is almost impossible. However, both models achieve relatively high Pass@1 scores on HumanEval. To assess whether these high scores may result from contamination, we conduct an analysis analogous to that in Table 1. The results are as follows (Table 10 in the revised paper):
>
> |                       | Portion | ROUGE | EM   |
> | :-------------------- | :------ | :---- | :--- |
> | Qwen2.5-7B            | 0.4     | 56.32 | 9.7  |
> |                       | 0.6     | 60.89 | 14.3 |
> |                       | 0.8     | 66.81 | 36.7 |
> | Llama-3.1-8B-Instruct | 0.4     | 23.42 | 0.0  |
> |                       | 0.6     | 32.13 | 1.2  |
> |                       | 0.8     | 47.32 | 8.4  |
>
> Accordingly, we categorize Qwen/HumanEval as red and Llama/HumanEval as green. In the revised version of the paper, we show Pass@K curves for LiveCodeBench in Figure 8. Neither model exhibits sufficiently strong capability on this benchmark; as K increases, their performance does not rise as sharply as it does on tasks such as Operation (shown in Figure 6). Consequently, we categorize Qwen/LiveCode and Llama/LiveCode as gray.
>
> For training, we use code-r1-12k [3] as the training data. Importantly, to keep the setup consistent with the math and logic tasks, we assign a correctness reward of 1 only when the generated code passes all test cases; otherwise, the model receives no correctness reward at all. Because code outputs are difficult to aggregate via majority voting, we do not evaluate the Vote setting in this task [4]. To reduce variance, we report avg@8, following the same protocol as SynLogic [5]. The results of the two code evaluation tasks are as follows (Table 11 in the revised paper):
>
> |                          | HumanEval    | LiveCodeBench |
> | ------------------------ | ------------ | ------------- |
> | **Qwen2.5-7B**           | **52.4**     | **6.2**       |
> | Standard                 | 65.2 (+12.8) | 15.4 (+9.2)   |
> | Format reward            | 58.5 (+6.1)  | 5.2 (-1.0)    |
> | Random reward            | 56.1 (+3.7)  | 5.0 (-1.2)    |
> | Incorrect Reward         | 53.0 (+0.6)  | 4.2 (-2.0)    |
> | EM                       | 58.1 (+5.7)  | 6.2 (0.0)     |
> | 1-shot (selected)        | 51.5 (-0.9)  | 6.0 (-0.2)    |
> | NSR                      | 55.1 (+2.7)  | 7.4 (+1.2)    |
> | **Llama3.1-8B-Instruct** | **68.4**     | **8.3**       |
> | Standard                 | 75.1 (+6.7)  | 12.0 (+3,7)   |
> | Format reward            | 69.2 (+0.8)  | 8.9 (+0.6)    |
> | Random reward            | 67.4 (-1.0)  | 7.3 (-1.0)    |
> | Incorrect Reward         | 68.8 (+0.4)  | 6.0 (-2.3)    |
> | EM                       | 70.4 (+2.0)  | 8.0 (-0.3)    |
> | 1-shot (selected)        | 69.0 (+0.6)  | 6.5 (-1.8)    |
> | NSR                      | 70.4 (+2.0)  | 6.8 (-1.5)    |
>
> From the table, we can see that under the standard RL setting, the model shows clear improvements on both HumanEval and LiveCodeBench. Under non-standard settings, only the red-category case—Qwen/HumanEval and the green-category case Llama/HumanEval—exhibit small gains, while no consistent improvement is observed in the weak-alignment setting. **This is consistent with our observations on math and logic tasks**. We appreciate your advice and will add this experiment to the main body of the paper in the next formal revision.

---

> ### Author Response · Authors · 2025-11-24
> **Response to Reviewer DMgH (2/3)**
>
> > **The paper equates model–task alignment with pass@k, but no prior work is cited to justify this choice of proxy. It would strengthen the argument to motivate pass@k more formally or to report an additional alignment signal (e.g. a divergence-based measure between base and trained models) as a complementary view.**
>
> Thanks for the suggestion. Here we report an additional alignment signal as advised. Specifically, we explore the KL divergence between the trained policy and the initial policy $\text{KL}(\pi_\theta||\pi_{ref})$ in three representative settings: Qwen on MATH500 (red, contamination and strong model-task alignment by pass@k), Qwen on Operation (green, strong model-task alignment by pass@k), and Qwen on SynLogic (gray, weak model-task alignment by pass@k). We use trained policies to generate trajectories via greedy decoding. Then, we feed these trajectories into the untrained reference model to obtain the log-probabilities for each token. We follow the approach used in DeepSeek-R1[6] for computing KL divergence:
> $$
> \text{KL}(\pi_\theta||\pi_{ref})=\frac{\pi_{ref}(o_i|q,o_{<i})}{\pi_{\theta}(o_i|q,o_{<i})}-\log \frac{\pi_{ref}(o_i|q,o_{<i})}{\pi_{\theta}(o_i|q,o_{<i})} - 1
> $$
> and compute the average KL over all tokens. The results are shown in the following table.
>
> |                   | Standard RL | Format | Random | Incorrect | Vote  | EM    | 1-shot | NSR   | PSR       |
> | ----------------- | ----------- | ------ | ------ | --------- | ----- | ----- | ------ | ----- | --------- |
> | MATH500 （red)    | 0.075       | 0.085  | 0.083  | 0.120     | 0.096 | 0.105 | 0.104  | 0.101 | 0.103     |
> | SynLogic (gray)   | **0.33**    | 0.052  | 0.074  | 0.145     | 0.102 | 0.119 | 0.102  | 0.082 | **0.312** |
> | Operation (green) | 0.127       | 0.120  | 0.090  | 0.150     | 0.130 | 0.132 | 0.12   | 0.100 | 0.089     |
>
> We also visualize it in the appendix of the revised paper (Figure 9). From the results, we observe that under the standard RL setting with correct reward, the divergence patterns correspond closely to model-task alignment measured by pass@k: the KL divergence is larger in the case of weak model-task alignment defined by pass@k, and smaller otherwise. For example, the distribution shift on SynLogic (weak model-task alignment by pass@k) before and after training is significantly larger than that observed on MATH500 and Operation. This suggests that, under the guidance of correct rewards, the model explores solutions that are substantially different from its initial policy, probably due to a weaker initial alignment with the SynLogic task. In contrast, for MATH500 and Operation, the model does not deviate markedly from its initial policy, which is already strong at the tasks from the beginning.
>
> It is worth noting that, in other RL settings without a correctness signal  or enough training samples (i.e., all the settings except standard RL and PSR), the KL divergence of all model-task pairs remains similar and small, which is probably because there are not strong enough signals to guide the model towards a direction stably in those settings.
>
> Therefore, our exploration here demonstrates that the divergence between the pre- and post-training output distributions in **a successful standard RL run** **with correct rewards** can serve as a complementary indicator of model–task alignment strength, where larger KL divergence corresponds to weaker model-task alignment. Its consistency with Pass@K further reinforces that Pass@K can be viewed as a reliable proxy for model–task alignment.

---

> ### Author Response · Authors · 2025-11-24
> **Response to Reviewer DMgH (3/3)**
>
> > **The set of weakly aligned models is fairly narrow. Including another family with weaker out-of-the-box performance on these** **tasks** **would make the alignment–effectiveness pattern more convincing.**
>
> We additionally include Mistral-7B-v0.1 in the weak-alignment setting. Its performance on math and logic tasks is worse than that of Llama-3.1-8B-Instruct, which allows us to provide more weakly aligned settings. Results are as follows (Table 12 in the revised paper):
>
> |                     | MATH         | AIME24     | AMC23       | SynLogic     | BBH          | BBEH       | OP           | CF           | Puzzle     | Logic        | Cipher     |
> | ------------------- | ------------ | ---------- | ----------- | ------------ | ------------ | ---------- | ------------ | ------------ | ---------- | ------------ | ---------- |
> | **Mistral-7B-v0.1** | **6.8**      | **0.0**    | **3.4**     | **0.0**      | **27.1**     | **0.4**    | **11.2**     | **22.8**     | **2.4**    | **28.4**     | **2.4**    |
> | GroundTruth         | 17.6 (+10.8) | 3.3 (+3.3) | 12.5 (+9.1) | 14.6 (+14.6) | 38.9 (+11.8) | 3.2(+2.8)  | 26.8 (+15.6) | 43.4 (+20.6) | 9.8 (+7.4) | 46.2 (+17.8) | 8.8 (+6.4) |
> | Format              | 6.4 (-0.4)   | 0.0 (0.0)  | 2.0 (-1.4)  | 0.0 (0.0)    | 24.3 (-2.8)  | 0.0 (-0.4) | 14.6 (+3.4)  | 26.2 (+3.4)  | 2.0 (-0.4) | 26.8 (-1.6)  | 3.2 (+0.8) |
> | Random              | 6.4 (-0.4)   | 0.0 (0.0)  | 3.2 (-0.2)  | 0.4 (+0.4)   | 21.2 (-5.9)  | 0.4 (0.0)  | 9.2(-2.0)    | 19.4 (-3.4)  | 2.0 (-0.4) | 22.4 (-6.0)  | 2.0 (-0.4) |
> | Incorrect           | 5.6 (-1.2)   | 0.0 (0.0)  | 1.0 (-2.4)  | 0.0 (0.0)    | 13.4 (-13.7) | 0.0 (-0.4) | 9.0 (-2.2)   | 19.2 (-3.6)  | 1.8 (-0.6) | 21.0 (-7.4)  | 1.8 (-0.6) |
> | Vote                | 5.8 (-1.0)   | 0.4 (+0.4) | 3.2 (-0.2)  | 0.0 (0.0)    | 14.2 (-12.9) | 0.4 (0.0)  | 12.8(+1.6)   | 28.6 (+5.8)  | 1.2 (-1.2) | 18.8 (-9.6)  | 3.6 (+1.2) |
> | EM                  | 7.8 (+1.0）  | 0.0 (0.0)  | 2.4 (-1.0)  | 0.8 (+0.8)   | 23.4 (-3.7)  | 0.4 (0.0)  | 10.4 (-0.8)  | 19.2 (-3.6)  | 1.4 (-1.0) | 24.4 (-4.0)  | 2.0 (-0.4) |
> | 1-shot (selected)   | 8.4 （+1.6） | 0.4 (+0.4) | 2.8 (-0.6)  | 0.0 (0.0)    | 22.4 (-4.7)  | 0.0 (-0.4) | 8.8 (-2.4)   | 19.8 (-3.0)  | 2.4 (0.0)  | 29.6 (+1.2)  | 2.0 (-0.4) |
> | NSR                 | 6.4 （-0.4） | 0.8 (+0.8) | 3.0 (-0.4)  | 0.0 (0.0)    | 24.2 (-2.9)  | 0.4 (0.0)  | 12.4 (+1.2)  | 23.2 (+0.4)  | 2.8 (+0.4) | 24.0 (-4.4)  | 2.0 (-0.4) |
>
> From the table above, we can see that Mistral shows stable improvements on math and logic tasks only under the standard RL setting. This further reinforces the validity of our conclusions under the weak-alignment setting.
>
> > **The exact criteria for each reward type and the RL setup used in that subsection should be made explicit, and the authors could briefly justify why they chose this particular RL recipe over newer GRPO-style variants (e.g., Dr.GRPO).**
> >
> > **The paper should state earlier and more prominently that DAPO is the default RL method used in the experiments.**
>
> Thank you very much for your suggestions on the writing. We have revised the paper accordingly and highlighted the changes in orange, including adjustments to the organization and clarifications of the methodology based on your comments.
>
> [1] Evaluating Large Language Models Trained on Code
>
> [2] LiveCodeBench: Holistic and Contamination Free Evaluation of Large Language Models for Code
>
> [3] Code-R1: Reproducing R1 for Code with Reliable Rewards
>
> [4] Spurious Rewards: Rethinking Training Signals in RLVR
>
> [5] SynLogic: Synthesizing Verifiable Reasoning Data at Scale for Learning Logical Reasoning and Beyond
>
> [6] DeepSeek-R1: Incentivizing Reasoning Capability in LLMs via Reinforcement Learning

---

### Meta-Review · Area_Chair_PZVT · 2026-01-07

**Summary:**

This paper was reviewed by four reviewers, with 3 positive recommendations and 1 negative commendations. The major concerns insufficient experiments, unclear clarifications. In the discussion period, the authors have provided additional experiments and more details. After carefully reading the comments and responses, the AC thinks that the major concerns have been well addressed.

**Reviewer Concerns:**

The major concerns have been well addressed.

**Reviewer Scores:**

Reviewer #DMgH may change the rating score to positive.

---

### Decision · Program_Chairs · 2026-01-26

Accept (Poster)